

# Control of inorganic and organic phosphorus molecules on microbial activity, and the stoichiometry of nutrient cycling in soils in an arid, agricultural ecosystem

Pamela Chavez-Ortiz[1,2], John Larsen[1], Gabriela Olmedo-Alvarez[3] and Felipe García-Oliva[1]

[1] Instituto de Investigaciones en Ecosistemas y Sustentabilidad, Universidad Nacional Autónoma de México, Morelia, Michoacán, Mexico
[2] Posgrado en Ciencias Biológicas, Universidad Nacional Autónoma de México, Mexico, Ciudad de México, Mexico
[3] Departamento de Ingeniería Genética, Centro de Investigación y de Estudios Avanzados del I.P.N., Irapuato, Guanajuato, Mexico

Corresponding author
Felipe García-Oliva,
fgarcia@cieco.unam.mx

## ABSTRACT

**Background**. The dynamics of carbon (C), nitrogen (N), and phosphorus (P) in soils determine their fertility and crop growth in agroecosystems. These dynamics depend on microbial metabolism, which in turn depends on nutrient availability. Farmers typically apply either mineral or organic fertilizers to increase the availability of nutrients in soils. Phosphorus, which usually limits plant growth, is one of the most applied nutrients. Our knowledge is limited regarding how different forms of P impact the ability of microbes in soils to produce the enzymes required to release nutrients, such as C, N and P from different substrates.

**Methods**. In this study, we used the arable layer of a calcareous soil obtained from an alfalfa cropland in Cuatro Cienegas, México, to perform an incubation experiment, where five different phosphate molecules were added as treatments substrates: three organic molecules (RNA, adenine monophosphate (AMP) and phytate) and two inorganic molecules (calcium phosphate and ammonium phosphate). Controls did not receive added phosphorus. We measured nutrient dynamics and soil microbial activity after 19 days of incubation.

**Results**. Different P molecules affected potential microbial C mineralization ($CO_2$-C) and enzyme activities, specifically in the organic treatments. P remained immobilized in the microbial biomass (Pmic) regardless of the source of P, suggesting that soil microorganisms were limited by phosphorus. Higher mineralization rates in soil amended with organic P compounds depleted dissolved organic carbon and increased nitrification. The C:N:P stoichiometry of the microbial biomass implied a change in the microbial community which affected the carbon use efficiency (CUE), threshold elemental ratio (TER), and homeostasis.

**Conclusion**. Different organic and inorganic sources of P affect soil microbial community structure and metabolism. This modifies the dynamics of soil C, N and P. These results highlight the importance of considering the composition of organic matter and

phosphate compounds used in agriculture since their impact on the microbial activity of the soil can also affect plant productivity.

# INTRODUCTION

In soil, orthophosphate anion ($HPO_4^{2-}$) produced from the weathering of apatite is the main source of inorganic phosphorus (P) available to the soil biota (*Paul, 2014*). However, this chemical form of P is not very abundant in the soil since it is very reactive and can generate different types of molecules through processes of precipitation, dissolution, and sorption (*Doolette, Smernik & Dougherty, 2011*). Another important source of P in the soil is organic P (*Turner, Cade-Menun & Westermann, 2003*), which is usually present in the form of inositol phosphates, such as phytate which can account for one-third to one-half of the total organic P in the soil (*Dalai, 1977*; *Gerke, 2015*; *Stewart & Tiessen, 1987*). Soil microbial can increase the availability of organic phosphorus molecules, and phosphates released by mineralization, through the action of secreted enzymes (exoenzymes) produced by soil microorganisms. For example, macromolecules, such as nucleic acids, can be depolymerized by the action of enzymes such as phosphodiesterases, or mineralized by phosphomonoesterases, phytases, and phosphonatases (*Paul, 2014*). Microorganisms can regulate their phosphorus demand in response to the availability of nutrients in the soil (*Tapia-Torres et al., 2015a*). However, the production of enzymes involved in the acquisition of P not only depends on the availability of the organophosphate substrate and inorganic phosphorus ($PO_4^{3-}$), but is also linked to the availability of carbon (C) (*Luo et al., 2019*), nitrogen (N) and other elements (*Olander & Vitousek, 2000*) such as magnesium (Mg) and calcium (Ca) (*Nannipieri et al., 2011*) and the presence of heavy metals (*Wiatrowska et al., 2015*). Microorganisms also produce enzymes that participate in the acquisition of C and N (ß-glucosidases and N-acetyl glucosaminidases, respectively) (*Sinsabaugh & Follstad Shah, 2012*).

Therefore, for microorganisms, the allocation of energy and nutrients for the production of enzymes and growth depends on the relative available quantities of these different elements, *i.e.,* the stoichiometry of elements in the microbial biomass and the availability of nutrients in the soil, or the relationships that exist between the essential elements C:N:P (*Elser & Sterner, 2002*; *Sinsabaugh, Carreiro & Repert, 2002*). The parameter *Threshold Elemental Ratio* (TER) can identify the C:N or C:P ratios at which microbial metabolism changes from being controlled by the supply of energy (C) to being controlled by the supply of nutrients such as N and P (*Sterner & Elser, 2002*; *Sinsabaugh & Follstad Shah, 2012*). TER analyses have been reported for natural terrestrial ecosystems (*Tapia-Torres et al., 2015a*; *Montiel-González et al., 2017*; *Cui et al., 2018a*; *Cui et al., 2018b*) and managed ecosystems (*Zhang et al., 2020*) but only a few studies have analyzed TER in agricultural systems (*Chávez-Ortiz et al., 2022*; *Cui et al., 2022*; *Zheng et al., 2020*). Ecological stoichiometric

analysis in agricultural systems is an important tool with which to better understand the effect of fertilizers on soil microbial communities and the coupling of nutrient cycles (*Chen et al., 2024*). This is valuable information in terms of practicing sustainable food production that can avoid the loss of soil microorganism diversity and thus maintain their provision of ecosystem services (*Van De Waal et al., 2018*).

Carbon use efficiency (CUE) represents the efficiency with which bacterial populations convert organic carbon substrates into biomass and is quantified as carbon accumulation in biomass (biomass production or sequestration) relative to carbon released from organic matter. CUE corresponds to the rate at which microbial communities decompose organic matter and release $CO_2$ (*Manzoni et al., 2012*; *Moorhead, Lashermes & Sinsabaugh, 2012*; *Sinsabaugh & Follstad Shah, 2012*) and is a function of the ability of the microbial community to regulate enzyme expression and biomass composition to reduce the difference between nutrients in environmental resources and growth requirements and enable a maximized growth rate (*Sinsabaugh & Follstad Shah, 2012*). Microbial CUE varies with environmental conditions, such as resource stoichiometry and availability and thus depends to a great extent on the composition of the organic matter (OM) and decreases when the OM is made up of recalcitrant compounds (*Sinsabaugh et al., 2013*). The chemical composition of composts and other organic soil amendments predict decomposition and nitrogen mineralization rates (*Rowell, Prescott & Preston, 2001*; *Flavel & Murphy, 2006*). *Rowell, Prescott & Preston (2001)* found that the alkyl group was highly positively correlated to N mineralization. Commonly, the presence of proteins is related to carboxylic and N-O-alkyl signals, including methoxyls, which are a particularly labile fraction of the organic pool, while the phenolic index, representing lignin or phenolic acids, was a factor that reduced N mineralization. However, *Flavel & Murphy (2006)* found no correlation with a specific group of the $^{13}$C-NMR spectra but found that N mineralization was positively related to initial the total C and N values of the amendments, as well as the cellulose and lignin content, while C mineralization was positively correlated to total C, cellulose and $NH_4^+$ concentrations. Other studies have focused on C:N ratios of organic amendments as an indicator of quality and complexity and assessed their effect on soil fertility (*Scotti et al., 2015a*; *Scotti et al., 2015b*; *Scotti et al., 2016*) as a predictor of C and N mineralization rates and N immobilization by microorganisms. *Hodge, Robinson & Fitter (2000)* and *Scotti et al. (2015a)*; *Scotti et al. (2015b)* reported that microbial growth can be limited by a C:N ratio of between 25–30 in organic soil amendments, promoting temporary N immobilization and impairing crop growth. Previous studies reported that the inorganic fertilization decreases both soil C:N:P ratio and soil microbial C:N:P ratios (*Zheng et al., 2021*; *Sun et al., 2022*), while the fertilization with organic amendments increases soil microbial C:P ratios (*Chen et al., 2024*), generating imbalanced soil C:N:P ratios (*Sun et al., 2022*). However, few studies have addressed the chemical composition of P molecules in fertilizers and organic amendments on microbial activity.

Since P is an important fertilizer applied in agricultural fields and an essential element for soil microorganisms, but also one that is dependent on C and N for its acquisition, in this study, we analyzed how different inorganic and organic phosphate compounds with different complexities and that can be found in fertilizers or organic matter, modify

the stoichiometry and microbial activity in the soil. We used agricultural soil from the Cuatro Cienegas Basin in Coahuila, Mexico (CCB), a desert characterized by its low phosphorus availability in the soil (*Tapia-Torres & García-Oliva, 2013*; *Tapia-Torres et al., 2016*). In these soils, microorganisms develop various adaptive phosphorous acquisition strategies, which are related to the production of exoenzymes, also known as ecoenzymes (*Tapia-Torres et al., 2016*).

In this study, we employed soil microcosms to evaluate the effects of the incorporation of some of the most common organic compounds found in organic matter (OM) on the transformation of nutrients and microbial activity in the soil. We evaluated inositol phosphates (phytic acid), nucleic acids in their macromolecular form (RNA), and a monophosphate ester, such as adenosine monophosphate (AMP), as well as the effects of inorganic P molecules commonly used in mineral fertilizers, such as monoammonium phosphate (MAP) and calcium phosphate. Applying the concepts of ecological stoichiometry (CUE and TER), we also determined how the different sources of P can modify nutrient limitations for microorganisms and the efficiency of carbon use. We hypothesized that labile organic P molecules (monoester phosphate AMP and diester phosphate RNA) can improve nutrient availability by stimulating microbial community activity. On the other hand, since phytic acid molecules can be a source of carbon, but not of N, we hypothesize that the effect of the phytic acid on the microbial community will depend on the capacity of microorganisms to produce phytases. We also hypothesized that the application of inorganic P (MAP and calcium phosphate) only benefits microbial communities until they become limited by either energy or.

## MATERIALS AND METHODS

### Study site

This study was carried out with samples obtained from an alfalfa crop plot located on the western side of the Cuatro Cienegas Basin (26°58′57″N, 102°5′10″W). The climate at Cuatro Cienegas Basin (CCB) is hot and arid, with an average yearly temperature of 21.9 °C and an average annual precipitation of 253 mm (*Montiel-González et al., 2021*). The dominant parent material in the west of CCB is calcium carbonate (*Mc Kee, Jones & Long, 1990*) and the dominant soil groups are Calcisols (*García-Oliva et al., 2018*), which is the soil group corresponding to the obtained samples.

Management at the farming plots consists of fertilization every 25 days, principally with MAP (monoammonium phosphate) technical grade, NPK 20−20−20 fertilizers, or NPK 11−42−0 fertilizers. Vermicompost leachate is often applied at a dose of 100 L ha$^{-1}$. Insecticides are used according to requirement and herbicides with the active compound clethodim are used to control grasses.

### Soil sampling

Soil sampling from the alfalfa crop plot was conducted in August 2018. We established a 50 × 50 m plot within the alfalfa crop. Soil samples were taken along five transects chosen randomly on one side of 50 m. A subsample was taken each 10 m along each transect, obtaining five subsamples that were mixed homogeneously to produce one composite

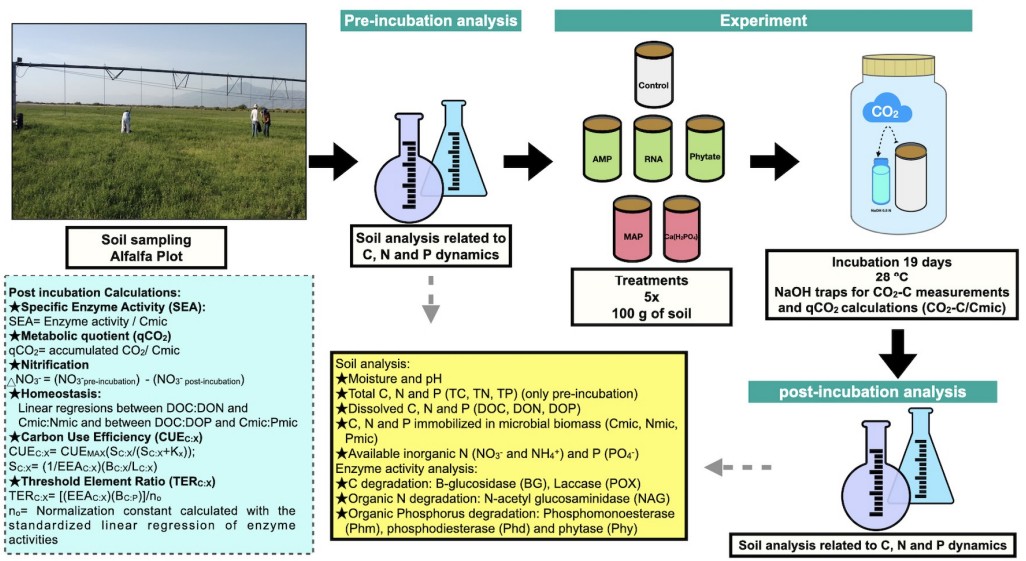

**Figure 1** **Methods summary.** Soil laboratory analysis are described in the yellow box, which is related to pre-incubation and post-incubation analysis. Calculations made from chemical and enzymatic variables are specified in the blue box. Photo and figures by Pamela Chávez-Ortíz.

soil sample per transect. Soil samples were taken from the top 15 cm of the mineral soil with a soil core sampler, placed in black plastic bags, and stored at 4 °C until subsequent laboratory analysis.

## Experimental design and incubation

An incubation experiment was conducted using soil amended with different phosphorus compounds (Fig. 1). The experimental design consisted of one factor, with six levels: five phosphorus compounds and one negative control. The added phosphorus compounds were chosen based on the most common organic P compounds found in farming soils, such as phosphate monoesters, phosphate diesters, and phytic acid. We used adenosine monophosphate (AMP) as a phosphate monoester, and torula yeast RNA (Sigma-Aldrich) as a phosphate diester. We also used two inorganic phosphate compounds: monoammonium phosphate (MAP; used at the study site as fertilizer) and monobasic calcium phosphate $(Ca(H_2PO_4)_2)$, known as triple superphosphate and commonly used for fertilizer production. The concentrations of the phosphorus compounds added to the soil were calculated according to a maximum P concentration used as a fertilizer in the sampled site (16.5 kgP ha$^{-1}$). A concentration of 27.8 μg P g$^{-1}$ of soil was added, which corresponds to 89.87 μmol P (Table 1), calculated based on the PVC area of 0.0019 m$^2$. Phosphorus sources were added to the water used to adjust the soil samples to water holding capacity.

We included five replicates for each treatment, corresponding to each composite soil sample obtained from the field. Soil incubations were carried out over 19 days, (time defined by the obtained C mineralization rate data) at 28 °C. A soil sample of 100 g was added to sterilized PVC tubes with one extreme closed with a mesh (pore size <0.05 mm).

**Table 1** Phosphorus content and concentration of each chemical compound added to the fertilization experiment.

|  | P content g mol$^{-1}$ on dry basis | Grams of reactant added per sample (25% water weight) | P added per sample (g) | μmol P per sample |
|---|---|---|---|---|
| Phytic acid sodium salt hydrate | 185.82 | 0.013 | 0.0028 | 90 |
| Adenosine monophosphate | 30.97 | 0.033 | 0.0028 | 90 |
| RNA from torula yeast | 9.8% | 0.035 | 0.0028 | 90 |
| Monoammonium phosphate | 30.97 | 0.0103 | 0.0028 | 90 |
| Calcium phosphate monobasic | 61.94 | 0.011 | 0.0028 | 90 |

Water was added to reach 90% of the field capacity. During the incubation period, soil water was maintained by weight measurement. PVC tubes were placed into 1 L glass flasks and sealed during the incubation, as shown in Fig. 1.

## Potential C mineralization

Carbon mineralization was measured periodically during the 19 days of incubation. For this analysis, $CO_2$ traps were placed inside the glass flasks. These traps consisted of a vial containing 10 ml of NaOH 1N, which was titrated periodically each third day with HCl 1N and $BaCl_2$ and replaced with fresh NaOH solution. The HCl used for titration was used to calculate C mineralization rates (Coleman et al., 1977). The metabolic quotient for $CO_2$ ($qCO_2$) was determined according to Anderson & Domsch (1993), dividing the accumulated $CO_2$-C by microbial biomass C following incubation (Cmic).

## Biogeochemical and enzymatic activity analyses

Before and after incubation, biogeochemical analysis and determination of enzymatic activities were performed. Soil moisture content was determined by gravimetric analysis, drying the samples at 100 °C until reaching constant weight. Active soil pH in deionized water (1:10 w/v) was measured using a digital potentiometer (Thermo Scientific Orion 3star Plus). The weight of the samples for all analyses was corrected with the fraction of dry soil obtained in the moisture content determination.

Total C, N, and P were quantified (TC, TN, and TP) using dry soil ground in an agate mortar. Total C (TC) and total inorganic C (TIC) were determined by coulometric detection (Huffman, 1977) in a total Carbon Analyzer (UIC model CM5012). Total organic C (COT) was calculated by the difference between TC and TIC. TN and TP were determined following acid digestion, in which TN was determined by the Kjeldahl macro method (Bremmer, 1996) and TP by the reduction of molybdate with ascorbic acid (Murphy & Riley, 1962). Both nutrients were measured by colorimetry in a Bran-Lubbe Auto Analyzer 3 (Norderstedt, Germany).

The available forms of nitrogen ($NH_4^+$ and $NO_3^-$) were extracted from 10 g of fresh soil with 2 M KCl, followed by filtration through a Watman No. 1 paper filter and determined by colorimetry with the phenol-hypochlorite method (Robertson et al., 1999). Available inorganic P ($HPO_4^{2-}$) was extracted from 5 g of fresh soil with 0.5M solution of $NaHCO_3$,

adjusted to pH 8.5 and determined colorimetrically by the molybdate-ascorbic acid method (*Murphy & Riley, 1962*; *Tiessen & Moir, 1993*).

The dissolved organic nutrients were determined by the difference between the total dissolved nutrient (C, N, or P) and the dissolved inorganic nutrient. Dissolved organic C, N, and P (DOC, DON, and DOP) were extracted with deionized water (1:4 w/v) according to *Jones & Willett (2006)* and filtered through a Millipore 0.45 $\mu$m filter. Filtrates were used directly to measure inorganic dissolved N and P, and total and inorganic dissolved C. For total dissolved N and P, the filtrate was acid digested. Total and inorganic forms of dissolved N and P were quantified in a Bran-Luebbe Auto analyzer 3 (Norderstedt, Germany). For determination of the DOC, the total dissolved C (TDC) and dissolved inorganic C (DIC) were measured in a Carbon Autoanalyzer (TOC CM 5012).

The amounts of C, N, and P within the microbial biomass (Cmic, Nmic, and Pmic) were obtained by the method of fumigation with chloroform and incubation for 24 h at 27 °C (*Vance, Brookes & Jenkinson, 1987*). Cmic and Nmic were extracted using 0.5 M $K_2SO_4$, according to *Brookes et al. (1985)*, and filtered with Whatman No. 42 and No. 1, respectively. Cmic was quantified using a Carbon Auto Analyzer (TOC CM 5012). C concentration was measured from each extract as total carbon (TCmic), using the module for liquids (UIC-COULOMETRICS), and as inorganic carbon (ICmic), using the acidification module CM 5130. For Nmic, the filtrate was acid digested and determined as TN by the Macro-Kjeldahl method (*Brookes et al., 1985*). The Pmic was extracted according to *Cole et al. (1977)*, using a solution of $NaHCO_3$ 0.5M and adjusted to pH 8.5, shaken for 16 h, and passed through Whatman No. 42 filters. Filtrates were digested using 11 N $H_2SO_4$ and a 50% w/v solution of ammonium persulfate and neutralized following the acid digestion. Microbial P was determined colorimetrically by the molybdate-ascorbic acid method using an Evolution 201 Thermo Scientific Inc. spectrophotometer at a wavelength of 880 nm (*Murphy & Riley, 1962*). Nutrients in microbial biomass were calculated by subtracting non-fumigated sample data from that of fumigated samples and then dividing by the corresponding conversion factor. kEC (0.45) and kEN (0.54), determined by *Joergensen (1996)* and *Joergensen & Mueller (1996)*, were used to calculate Cmic and Nmic, respectively, and a *Kp* correction factor of 0.4 (*Hedley, Stewart & Chauhan, 1982*; *Lajtha & Jarrell, 1999*) was used for the Pmic calculations.

The differences ($\Delta$) between biogeochemical variables before and after incubation were calculated by subtracting the values at the beginning of the incubation from those at the end. Net nitrification was therefore calculated by subtracting the values of available $NO_3^-$ after incubation from those of available $NO_3^-$ before incubation.

The enzymatic activities of phosphomonoesterase (Phm), phosphodiesterase (Phd), phytase (Phy) beta-glucosidase (BG), N-acetyl glucosaminidase (NAG), and polyphenol oxidase (POX) were quantified. For these analyses, 2 g of fresh soil and 30 ml of modified universal buffer (MUB) at pH 8 were used for the ecoenzyme extraction. Three replicates and one control (sample with no substrate) were prepared per sample. Three substrate controls (substrate with no sample) were also included per assay, and all were incubated at 30 °C. The tubes were centrifuged after the incubation period and 750 $\mu$l of the supernatant was then diluted in 2 ml of deionized water and 75 $\mu$l NaOH 1N.

Measurements of the enzymatic activity of Phm, Phd, BG, and NAG are based on the spectrophotometric determination of p-nitrophenol (pNP) released from substrates linked to pNP, per unit of time ($\mu$mol pNP [g SDW]$^{-1}$ h$^{-1}$; *Tabatabai & Bremner, 1969*; *Verchot & Borelli, 2005*; *Fioretto et al., 2009*) and measured at 410 nm on an Evolution 201 spectrophotometer (Thermo Fisher Scientific, Waltham, MA, USA). The POX activity was determined by oxidation of the substrate 2,2′-azinobis-(-3 ethylbenzothiazoline-6-sulfononic acid) diammonium salt (ABTS), which was measured directly at a wavelength of 460 nm. Phy was quantified according to the method of phosphonatase enzymatic activity measurement described by *Tapia-Torres et al. (2016)*, using phytic acid as a substrate and quantifying the released Pi using the ascorbic acid reduction method (*Murphy & Riley, 1962*) measured at a wavelength of 882 nm. Phy activity was expressed as micromoles of inorganic P released per gram of soil dry weight per hour ($\mu$mol Pi [g SDW]$^{-1}$ h$^{-1}$).

Specific enzyme activity (SEA) was calculated to determine how much enzyme is synthesized per concentration of nutrients immobilized in microbial biomass. SEA was calculated according to *Waldrop, Balser & Firestone (2000)* and *Chávez-Vergara et al. (2016)*:

$$SEA = \text{Enzymatic activity/Carbon in microbial biomass}$$

where enzymatic activity is expressed in units of $\mu$mol g SDW$^{-1}$h$^{-1}$ and C in microbial biomass is expressed in units of mg C g SDW$^{-1}$.

## Homeostasis and the threshold element ratio (TER)

With the biogeochemical and enzymatic results obtained from the incubation experiment where different phosphorus compounds were applied to agricultural soil samples from the CCB, a homeostasis analysis was performed, performing simple linear regressions between the natural logarithm of DOC:DOP and the natural logarithm of Cmic:Pmic for C:P, and between the natural logarithm of DOC:DON and the natural logarithm of Cmic:Nmic for C:N. Taking the linear regression, it was assessed whether the slope differed from 0, which would mean a non-homeostatic microbial community. The elemental ratio thresholds (TER) were calculated in relation to the elements C:P (TER) and C:N (TER$_{C:N}$) according to *Sinsabaugh, Hill & Foolstad Shah (2009)*, using the following equations:

$$TERc:p = ((BG/(Phm+Phd))Bc:p)/\rho o \qquad (1)$$

$$TERC:N = ((BG/NAG)BC:N)/no \qquad (2)$$

where TER$_{C:P}$ is the threshold elemental ratio for elements C and P; BG/(Phm+Phd) is the ratio of enzymatic activity for B-1,4-glucosidase (BG) and the sum of phosphomonoesterase plus phosphodiesterase (Phm+Phd); BC: P is the C:P ratio for microbial biomass (Cmic/Pmic) and $\rho o$ is a normalization constant. For elements C and N, the TER$_{C:N}$ Eq. (2) is the threshold elemental ratio (dimensionless), (BG/(NAG)) is the ratio of enzymatic activity for $\beta$-glucosidase (BG) and N-acetyl glucosaminidase (NAG), B$_{C:N}$ is the C:N ratio for microbial biomass (Cmic:Nmic), and n$_o$ is a normalization constant.

The normalization constants are the intercept calculated with a standardized major axis regression type II (SMATR). For the constant $\rho o$, the regression is performed between the natural logarithms of the BG enzyme and the sum of the enzymes Phm and Phd. In contrast, for the constant $n_o$, the regression is calculated between the natural logarithms of the BG enzyme and the NAG enzyme. Equation (1) is modified from *Sinsabaugh, Hill & Foolstad Shah (2009)* since only the Phm enzyme is used in the original equation; however, the Phd enzyme has been included because of its importance and high activity in the soils of Cuatro Cienegas Basin (*Tapia-Torres et al., 2016*). The $TER_{C:P}$ and $TER_{C:N}$ results, converted to natural logarithms, were compared with the resource ratios (soil nutrients, DOC:DOP, and DOC:DON) using a Student's $t$-test. This can reveal whether the soil microorganisms are limited by energy (carbon) or by nutrients (N or P).

## Carbon use efficiency

Carbon use efficiency (CUE), in relation to N and P ($CUE_{C:N}$ and $CUE_{C:P}$), was calculated using the formulas developed in *Sinsabaugh et al. (2013)* and *Sinsabaugh et al. (2016)*:

$$CUEc:x = CUEMAX(Sc:x/(Sc:x+Kx)) \tag{3}$$

where X represents element N or P; $K_x$ is the mean saturation constant, which has a value of 0.5; $CUE_{MAX}$ is the upper limit for the efficiency of microbial growth, which has a value of 0.6 based on thermodynamic constraints; and $S_{C:X}$ is calculated as follows:

$$Sc:x = (1/(EEAc:x))(Bc:Lc:x) \tag{4}$$

where $EEA_{C:X}$ is the ratio of the enzymatic activities related to the nutrients C:X; $L_{C:X}$ is the ratio of the substrates consumed, which in this case were the dissolved organic nutrients of the soil, and $B_{C:X}$ the ratio of elements in microbial biomass. From the data obtained from $CUE_{C:P}$ and $CUE_{C:N}$, the CUE calculation was performed using the formula suggested by *Sinsabaugh & Follstad Shah (2012)* and *Sinsabaugh et al. (2016)*, as a best estimate for the CUE of the microbial community:

$$CUE = \sqrt{CUEC:NXCUEC:P} \tag{5}$$

## Statistical analysis

A one-way ANOVA was performed to determine the effect of the treatment on C mineralization, and on the biogeochemical and enzymatic variables, as well as on the differences between the beginning and the end of the incubation for enzyme activities and DOC, DON, DOP, $NO_3^-$, $PO_4^-$, $NH_4^+$, Cmic, Pmic, and Nmic. Residual frequency distribution was assessed with a Kruskal-Wallis test to verify a normal distribution (*García-Oliva & Maass, 1998*). A Tukey HSD test was performed after the ANOVA to identify differences between treatments. An ANOVA was also performed for the results of SEA, and an LSD test was performed after the ANOVA for the SEA obtained with Cmic. A Pearson correlation was performed among post-incubation biogeochemical variables, enzymatic activities (post-incubation), accumulated C mineralization, $qCO_2$, and nitrification. Principal component analysis (PCA) was conducted to determine which

variables explained variance in the results and to visualize the grouping of the different treatments. The data matrix was constructed using all biogeochemical and enzymatic data from all the samples, apart from those of SEA and $qCO_2$. The analysis was carried out using the function "prcomp" on R software. All statistical analyses were performed using R software (*R Core Team, 2020*). Given that three separate groups were observed in the PCA, Pearson correlation tests were conducted separately for each group.

One-way ANOVAs were performed to compare the results of $TER_{C:P}$, $TER_{C:N}$ (using the natural logarithm of TER), $CUE_{C:P}$, $CUE_{C:N}$, and CUE between treatments. Residual frequency distribution was assessed with a Kruskal–Wallis test to verify a normal distribution (*García-Oliva & Maass, 1998*). The Tukey HSD test was performed to identify the treatments with significant differences, except for the analysis conducted for the CUE in which no results were obtained with the Tukey HSD test, and an LSD analysis was performed. Student's $t$-tests were performed to identify differences between the TER values and the ratios between the dissolved organic nutrients. For the TER calculation, type II linear regressions were performed among the enzyme activities using the SMATR package. All statistical analyses were performed using R software (*R Core Team, 2020*).

## RESULTS

### Incubation experiment and metabolic quotient for $CO_2$ ($qCO_2$)

After 19 days of incubation, adenosine monophosphate (AMP) and RNA additions of P organic treatments presented the highest C mineralization (950 and 863 μg $CO_2$-C $g^{-1}$, respectively), while the $Ca(H_2PO_4)_2$ addition and the control treatments had the lowest C mineralization values (781 and 739 μg $CO_2$-C $g^{-1}$, respectively; Table 2). Cmic was lowest for the phytic acid treatment and therefore the calculated $qCO_2$ was higher for this phosphate ester treatment ($1.9 \pm 0.56$) compared to the control ($0.59 \pm 0.03$), suggesting a lower metabolic efficiency of the soil microbial community fertilized with phytate (Table 2). Cmic values in MAP, $Ca(H_2PO_4)2$, RNA and AMP were lower after the incubation compared to the Cmic values obtained before incubation (Table 3).

### Post-incubation biogeochemical analysis: changes in organic and inorganic C, N, and P pools and microbial P immobilization

AMP and RNA additions in P organic treatments produced higher $NO_3^-$ concentrations than the other treatments, as well as nitrification (Table 2). In contrast, $NH_4^+$ and $HPO_4^{2-}$ presented no significant differences between treatments (Table 2). Dissolved organic C (DOC) was significantly greater for the treatment with $Ca(H_2PO_4)_2$ than for those with RNA and AMP (Table 2). Moreover, the $Ca(H_2PO_4)_2$ andRNA treatments had higher dissolved organic N (DON) concentrations than the other treatments (Table 2).

Both organic (AMP and RNA) and inorganic $Ca(H_2PO_4)_2$ treatments increased dissolved organic P (DOP) (Table 2). Therefore, the control samples had higher DOC:DON and DOC:DOP ratios than the AMP and RNA treatments (Table 4). In addition, the control had a higher DON:DOP ratio than the monoammonium phosphate (MAP), AMP, RNA, and phytic acid treatments (Table 4). The control, MAP, and $Ca(H_2PO_4)_2$ treatments presented higher Cmic concentrations than the AMP, RNA, and phytic acid treatments (Table 2).

Chavez-Ortiz et al. (2024), *PeerJ*, DOI 10.7717/peerj.18140

Peer**J**

**Table 2  Data obtained for the different treatments and the control after 19 days incubation.** Data are means of pH and biogeochemical variables after incubation per treatment.

| Variable | Control | MAP (Pi) | Ca(H$_2$PO$_4$)$_2$ (Pi) | RNA (Po) | AMP (Po) | Phytic acid (Po) | F |
|---|---|---|---|---|---|---|---|
| pH (H$_2$O 1:5) | 8.1 ($\pm$0.12) | 8.1 ($\pm$0.060) | 8.1 ($\pm$0.050) | 8.2 ($\pm$0.040) | 8.1 ($\pm$0.030) | 8.1 ($\pm$0.06) | 0.66 |
| Cmic ($\mu$g g$^{-1}$) | 1,283 ($\pm$76)$^A$ | 1,027 ($\pm$69)$^{AB}$ | 1,057 ($\pm$78)$^{AB}$ | 673 ($\pm$99)$^{BC}$ | 611 ($\pm$93)$^{BC}$ | 690 ($\pm$13)$^C$ | 8.3$^{***}$ |
| Nmic ($\mu$g g$^{-1}$) | 45 ($\pm$9) | 39 ($\pm$4) | 43 ($\pm$6.000) | 58 ($\pm$6) | 46 ($\pm$5) | 50 ($\pm$12) | 0.82 |
| Pmic ($\mu$g g$^{-1}$) | 4.56 ($\pm$0.79)$^D$ | 15.44 ($\pm$3)$^{BCD}$ | 20.01 ($\pm$4.470)$^{BC}$ | 36.95 ($\pm$5)$^A$ | 27.22 ($\pm$1)$^{AB}$ | 10 ($\pm$2)$^{CD}$ | 13.5$^{***}$ |
| DOC ($\mu$g g$^{-1}$) | 164 ($\pm$24)$^{AB}$ | 95 ($\pm$12)$^{ABC}$ | 189 ($\pm$67)$^A$ | 50 ($\pm$6)$^{BC}$ | 30 ($\pm$4)$^C$ | 64 ($\pm$5)$^{ABC}$ | 4.7$^{***}$ |
| DON ($\mu$g g$^{-1}$) | 0.74 ($\pm$0.15)$^C$ | 0.57 ($\pm$0.120)$^C$ | 2.81 ($\pm$0.080)$^A$ | 2.17 ($\pm$0.160)$^B$ | 0.68 ($\pm$0.160)$^C$ | 0.69 ($\pm$0.06)$^C$ | 57.2$^{***}$ |
| DOP ($\mu$g g$^{-1}$) | 0.13 ($\pm$0.02)$^B$ | 0.38 ($\pm$0.060)$^{AB}$ | 0.57 ($\pm$0.110)$^A$ | 0.72 ($\pm$0.070)$^A$ | 0.7 ($\pm$0.080)$^A$ | 0.5 ($\pm$0.12)$^{AB}$ | 7.1$^{***}$ |
| NH$_4$ ($\mu$g g$^{-1}$) | 0 ($\pm$0) | 0.004 ($\pm$0.004) | 0 ($\pm$0) | 0.062 ($\pm$0.062) | 0 ($\pm$0) | 0 ($\pm$0) | 0.97 |
| NO$_3$ ($\mu$g g$^{-1}$) | 57 ($\pm$6)$^C$ | 69 ($\pm$9)$^C$ | 51 ($\pm$6)$^C$ | 102 ($\pm$7)$^B$ | 135 ($\pm$7)$^A$ | 58 ($\pm$7)$^C$ | 22.4$^{***}$ |
| HPO$_4$ ($\mu$g g$^{-1}$) | 6.3 ($\pm$0.96) | 15.2 ($\pm$2) | 17.3 ($\pm$5) | 10.6 ($\pm$1.43) | 11.9 ($\pm$1.35) | 10.8 ($\pm$1) | 2.3 |
| Nitrification ($\mu$gNO$_3$ g$^{-1}$) | 37 ($\pm$6)$^C$ | 54 ($\pm$8)$^C$ | 36 ($\pm$5)$^C$ | 87 ($\pm$7)$^B$ | 120 ($\pm$6)$^A$ | 43 ($\pm$6)$^C$ | 29.4$^{***}$ |
| CO$_2$-C ($\mu$gCO$_2$-C g$^{-1}$) | 739 ($\pm$6)$^D$ | 831 ($\pm$11)$^{BC}$ | 781 ($\pm$17)$^{CD}$ | 863 ($\pm$9)$^B$ | 950 ($\pm$17)$^A$ | 802 ($\pm$4)$^C$ | 39.2$^{***}$ |
| qCO$_2$ | 0.59 ($\pm$0.032)$^B$ | 0.82 ($\pm$0.049)$^{AB}$ | 0.83 ($\pm$0.084)$^{AB}$ | 1.52 ($\pm$0.244)$^{AB}$ | 1.60 ($\pm$0.216)$^{AB}$ | 1.9 ($\pm$0.55)$^A$ | 4$^{**}$ |

**Notes.**

AMP, Adenosine monophosphate; MAP, Monoammonium phosphate; DOC, Dissolved organic carbon; DON, Dissolved Organic Nitrogen; DOP, Dissolved organic phosphorus; NH4, Available ammonium; NO3, Available nitrate; HPO4, Available inorganic phosphate; Cmic, Carbon immobilized in microbial biomass; Nmic, Nitrogen immobilized in microbial biomass; CO2-C, Carbon from CO2 produced in mineralization; q-CO2, Metabolic quotient; Pi, Inorganic phosphorus source; Po, Organic phosphorus source.

Standard error in parentheses. Letters (A, B, C, D) show significant differences between treatments.

$^*p < 0.05$.
$^{**}p < 0.01$.
$^{***}p < 0.001$.

Mean and standard error are displayed for $n = 5$ for each treatment.

**Table 3** Soil p hysic and chemical, biogeochemical and enzymatic activity values obtained from the alfalfa crop soil before the incubation fertilization experiment.

| Variable | Mean (Stándard error) |
|---|---|
| Soil moisture (%) | 29 ($\pm$0.006) |
| pH in water | 8.1 ($\pm$0.017) |
| TOC (mg g$^{-1}$) | 26 ($\pm$1.2) |
| TN (mg g$^{-1}$) | 2.6 ($\pm$0.15) |
| TP (mg g$^{-1}$) | 0.6 ($\pm$0.045) |
| DOC ($\mu$g g$^{-1}$) | 28 ($\pm$2.9) |
| DON ($\mu$g g$^{-1}$) | 1.64 ($\pm$0.51) |
| DOP ($\mu$g g$^{-1}$) | 0.28 ($\pm$0.084) |
| NH$_4$ ($\mu$g g$^{-1}$) | 0.22 ($\pm$0.216) |
| NO$_3$ ($\mu$g g$^{-1}$) | 15 ($\pm$1.5) |
| HPO$_4$ ($\mu$g g$^{-1}$) | 3.4 ($\pm$1.479) |
| Cmic ($\mu$g g$^{-1}$) | 1,190 ($\pm$192) |
| Nmic ($\mu$g g$^{-1}$) | 70 ($\pm$5.4) |
| Pmic ($\mu$g g$^{-1}$) | 5 ($\pm$0.86) |
| Cmic: Nmic | 17 ($\pm$3) |
| Cmic: Pmic | 246 ($\pm$33) |
| Nmic: Pmic | 16 ($\pm$3) |
| Phm ($\mu$mol pNP [g SDW]$^{-1}$ h$^{-1}$) | 0.030 ($\pm$0.015) |
| Phd ($\mu$mol pNP [g SDW]$^{-1}$ h$^{-1}$) | 0.157 ($\pm$0.041) |
| Phy ($\mu$mol Pi [g SDW]$^{-1}$ h$^{-1}$) | 0.056 ($\pm$0.056) |
| NAG ($\mu$mol pNP [g SDW]$^{-1}$ h$^{-1}$) | 0.008 ($\pm$0.003) |
| BG ($\mu$mol pNP [g SDW]$^{-1}$ h$^{-1}$) | 0.005 ($\pm$0.002) |
| POX ($\mu$mol tyrosine [g SDW]$^{-1}$ h$^{-1}$) | 0.196 ($\pm$0.046) |

**Notes.**

TOC, Total organic carbon; TN, Total nitrogen; TP, Total phosphorus; DOC, Dissolved organic carbon; DON, Dissolved Organic Nitrogen; DOP, Dissolved organic phosphorus; NH4, Available ammonium; NO3, Available nitrate; HPO4, Available inorganic phosphate; Cmic, Carbon immobilized in microbial biomass; Nmic, Nitrogen immobilized in microbial biomass; Pmic, Phosphorus immobilized in microbial biomass; Phm, Phosphomonoesterase enzyme activity; Phd, Phosphodiesterase enzyme activity; Phy, Phytase enzyme activity; NAG, N-acetyl glucosaminidase enzyme activity; BG, $\beta$-glucosidase enzyme activity; POX, Polyphenol oxidase (laccase) enzyme activity.
Standard error values are shown inside parenthesis. Mean and standard error are displayed for $n = 5$ for each treatment.

In contrast, treatments with RNA, AMP, and Ca(H$_2$PO$_4$)$_2$ immobilized significantly more P compared to the control treatment (Table 3). The organic treatments favored N immobilization in microbial biomass given that the Cmic:Nmic ratio was lower in these P organic treatments (RNA, AMP, and phytic acid) than in the control treatment (Table 4). These results suggest that the organic P treatments favored P and N immobilization in microbial biomass (Pmic) and high dissolved organic P (DOP) as well as higher available nitrate.

## Enzyme and specific enzyme activity

Enzyme activity was only significantly higher for NAG, where in the Ca(H$_2$PO$_4$)$_2$ treatment (Table S1). The specific enzyme activity obtained by enzyme activity normalization using Cmic differed significantly among the enzymes POX, NAG, and Phd (Table 5).

**Table 4  Data for nutrient immobilization obtained from dissolved nutrients and biomass ratios.** Means of dissolved nutrient ratios and nutrient within microbial biomass ratios, which denotes higher nutrient immobilization in microbial biomass the lower the value of the ratio.

| Variable | C | MAP (Pi) | $Ca(H_2PO_4)_2$ (Pi) | RNA (Po) | AMP (Po) | Phytic acid (Po) | F |
|---|---|---|---|---|---|---|---|
| DOC:DON | $295^A$ ($\pm99$) | $225^{AB}$ ($\pm84$) | $68^{AB}$ ($\pm25$) | $24^B$ ($\pm3.80$) | $52^B$ ($\pm10$) | $95^{AB}$ ($\pm6.0$) | $3.4^{**}$ |
| DOC:DOP | $1{,}792^A$ ($\pm720$) | $312^{AB}$ ($\pm94$) | $367^{AB}$ ($\pm126$) | $74^B$ ($\pm12$) | $43^B$ ($\pm6.8$) | $193^B$ ($\pm82$) | $4.1^{**}$ |
| DON:DOP | $8.1^A$ ($\pm3.5$) | $1.5^B$ ($\pm0.14$) | $5.7^{AB}$ ($\pm1.20$) | $3.1^B$ ($\pm0.31$) | $0.816^B$ ($\pm0.16$) | $1.9^B$ ($\pm0.66$) | $2.7^*$ |
| Cmic:Pmic | $310^A$ ($\pm44$) | $82^B$ ($\pm18$) | $68^B$ ($\pm20$) | $18^B$ ($\pm1.20$) | $25^B$ ($\pm3.3$) | $58^B$ ($\pm3.8$) | $25.8^{***}$ |
| Cmic:Nmic | $34^A$ ($\pm7.4$) | $28^{AB}$ ($\pm4.3$) | $26^{ABC}$ ($\pm2.9$) | $11^C$ ($\pm0.99$) | $15^{BC}$ ($\pm0.93$) | $14^{BC}$ ($\pm2.1$) | $5.6^{***}$ |
| Nmic:Pmic | $10^A$ ($\pm2.7$) | $2.9^B$ ($\pm0.55$) | $2.5^B$ ($\pm0.41$) | $1.66^B$ ($\pm0.20$) | $1.7^B$ ($\pm0.16$) | $4.6^B$ ($\pm0.48$) | $8.7^{***}$ |

**Notes.**

Cmic: Nmic and Cmic: Pmic. Standard error is shown in parenthesis. Letters (A, B, C, D) indicate significant differences between treatments with the Tukey test.

AMP, adenosine monophosphate; MAP, monoammonium phosphate.

$^*p < 0.05$

$^{**}p < 0.01$

$^{***}p < 0.001$

Mean and standard error are displayed for $n = 5$ for each treatment.

The RNA and phytic acid treatments had higher POX SEA than the inorganic P treatments (MAP and $Ca(H_2PO_4)_2$). Organic P treatments also produced higher Phd SEA than the inorganic P treatments and the control (Table 5) suggesting that the microbes used phosphodiesterase to obtain P from these substrates.

The phytic acid-treated samples had the highest NAG SEA values while the RNA, AMP, and control treatments had the lowest (Table 5). N-acetyl glucosaminidase is one of three enzymes that catalyze the hydrolysis of chitin, which is important in carbon (C) and nitrogen (N) cycling in soils. It participates in chitin conversion to amino sugars, which are major sources of mineralizable N in soils.

## Increases in DOC, DOP, and Pmic after the incubation experiment

The $Ca(H_2PO_4)_2$ and the AMP treatments presented the highest and lowest increases ($\Delta$) in DOC concentration after incubation, respectively (Table 6). Similarly, the $Ca(H_2PO_4)_2$ treatment had the highest DON increase, but the lowest increase was in the MAP treatment. In contrast, the $Ca(H_2PO_4)_2$, RNA, and AMP treatments had higher increases in DOP than the control, which had negative values (Table 6).

Among microbial nutrients, only Pmic presented a significant increase after incubation. Among treatments, the RNA and control presented the highest and lowest values, respectively (Table 6).

## A complex dynamic observed from the application of inorganic and organic P fertilization

In the PCA, the first and second components explained 26% (eigenvalue = 3.69) and 17% (eigenvalue = 2.32) of variance, respectively (Table S1). $NO_3^-$ and the NAG enzyme were the variables with greater weight in the first component, while $HPO_4^{2-}$ and the POX enzyme better explained the variance of the second component (Fig. 2). The treatments clustered into three groups: only control samples on the left side of the first component and negative values of the second component; the $Ca(H_2PO_4)_2$, MAP and phytic acid-treated

Chavez-Ortiz et al. (2024), *PeerJ*, DOI 10.7717/peerj.18140

Peer J

**Table 5  Specific Enzyme activities (SEA) after 19 days incubation.** Means of specific enzyme activities per treatment, obtained with the division between enzymatic activity and Cmic.

| Variable | Control | MAP (Pi) | Ca(H$_2$PO$_4$)$_2$ (Pi) | RNA (Po) | AMP (Po) | Phytic acid (Po) | F |
|---|---|---|---|---|---|---|---|
| SEA BG ($\mu$mol pNP mgCmic$^{-1}$ h$^{-1}$) | 0.069 ($\pm$0.02) | 0.06 ($\pm$0.014) | 0.049 ($\pm$0.012) | 0.102 ($\pm$0.038) | 0.139 ($\pm$0.057) | 0.132 ($\pm$0.074) | 0.8 |
| SEA POX ($\mu$mol tyr mgCmic$^{-1}$ h$^{-1}$) | 0.244 ($\pm$0.04)$^{AB}$ | 0.171 ($\pm$0.050)$^{B}$ | 0.082 ($\pm$0.052)$^{B}$ | 0.493 ($\pm$0.082)$^{A}$ | 0.371 ($\pm$0.195)$^{AB}$ | 0.540 ($\pm$0.134)$^{A}$ | 2.9* |
| SEA NAG ($\mu$mol pNP mgCmic$^{-1}$ h$^{-1}$) | 0.014 ($\pm$0.006)$^{B}$ | 0.027 ($\pm$0.006)$^{AB}$ | 0.030 ($\pm$0.007)$^{AB}$ | 0.004 ($\pm$0.001)$^{B}$ | 0.001 ($\pm$0.001)$^{B}$ | 0.066 ($\pm$0.033)$^{A}$ | 2.8* |
| SEA Phm ($\mu$mol pNP mgCmic$^{-1}$ h$^{-1}$) | 0.013($\pm$0.007) | 0.025 ($\pm$0.015) | 0.16 ($\pm$0.134) | 0.068 ($\pm$0.017) | 0.097 ($\pm$0.054) | 0.052 ($\pm$0.017) | 0.8 |
| SEA Phd ($\mu$mol pNP mgCmic$^{-1}$ h$^{-1}$) | 0.164 ($\pm$0.01)$^{B}$ | 0.171 ($\pm$0.018)$^{B}$ | 0.167 ($\pm$0.015)$^{B}$ | 0.429 ($\pm$0.123)$^{A}$ | 0.408 ($\pm$0.078)$^{A}$ | 0.480 ($\pm$0.13)$^{A}$ | 3.5* |
| SEA Phy ($\mu$mol Pi mgCmic$^{-1}$ h$^{-1}$) | 0.896 ($\pm$0.31) | 0.403 ($\pm$0.207) | 0.793 ($\pm$0.415) | 0.696 ($\pm$0.373) | 1.213 ($\pm$0.61) | 0.497 ($\pm$0.32) | 0.55 |

**Notes.**

The standard error is indicated between parentheses. Letters (A, B, C, D) indicate significant differences between treatments as obtained from the Tukey test.

AMP, adenosine monophosphate; MAP, monoammonium phosphate; BG, $\beta$ glucosidase; NAG, N-acetyl glucosaminidase; POX, Polyphenol oxidase (laccase); Phm, Phosphomonoesterase; Phd, Phosphodiesterase; Phy, Phytase.

*$p < 0.05$
**$p < 0.01$
***$p < 0.001$
Mean and standard error are displayed for $n = 5$ for each treatment.

**Table 6  Dissolved organic nutrients (DOC, DON and DOP) and nutrients immobilized in microbial biomass (Cmic, Nmic and Pmic) after organic or inorganic P fertilization treatment.** Means of the differences (D) between post and pre incubation values for dissolved organic nutrients (DOC, DON and DOP) and nutrients immobilized in microbial biomass (Cmic, Nmic and Pmic).

| Variable | Control | MAP (Pi) | $Ca(H_2PO_4)_2$ (Pi) | RNA (Po) | AMP (Po) | Phytic acid (Po) | F |
|---|---|---|---|---|---|---|---|
| $\Delta$DOC ($\mu$g g$^{-1}$) | 136 ($\pm$24)$^{AB}$ | 67 ($\pm$13)$^{ABC}$ | 160 ($\pm$67)$^{A}$ | 22 ($\pm$4.5)$^{BC}$ | 1.5 ($\pm$2.1)$^{C}$ | 36 ($\pm$7.7)$^{ABC}$ | 4.7** |
| $\Delta$DON ($\mu$g g$^{-1}$) | −0.90 ($\pm$0.45)$^{BC}$ | −1.07 ($\pm$0.59)$^{C}$ | 1.17 ($\pm$0.54)$^{A}$ | 0.53 ($\pm$0.48)$^{AB}$ | −0.96 ($\pm$0.64)$^{BC}$ | −0.95 ($\pm$0.55)$^{BC}$ | 3.12* |
| $\Delta$DOP ($\mu$g g$^{-1}$) | −0.15 ($\pm$0.07)$^{B}$ | 0.06 ($\pm$0.14)$^{AB}$ | 0.30 ($\pm$0.12)$^{A}$ | 0.45 ($\pm$0.08)$^{A}$ | 0.41 ($\pm$0.06)$^{A}$ | 0.23 ($\pm$0.14)$^{AB}$ | 5** |
| $\Delta$Cmic ($\mu$g g$^{-1}$) | 94 ($\pm$218) | −163 ($\pm$217) | −133 ($\pm$154) | −517 ($\pm$254) | −500 ($\pm$228) | 32 ($\pm$278) | 1.4 |
| $\Delta$Nmic ($\mu$g g$^{-1}$) | −26 ($\pm$12) | −32 ($\pm$6.4) | −27 ($\pm$7.3) | −12 ($\pm$10) | −25 ($\pm$7.1) | −0.15 ($\pm$16) | 0.43 |
| $\Delta$Pmic ($\mu$g g$^{-1}$) | −0.48 ($\pm$1.4)$^{D}$ | 10 ($\pm$3.5)$^{BCD}$ | 15 ($\pm$4.9)$^{BC}$ | 32 ($\pm$5.2)$^{A}$ | 22 ($\pm$1.1)$^{AB}$ | 5.5 ($\pm$2.1)$^{CD}$ | 11.6** |

**Notes.**

The standard error is indicated between parentheses. AMP, adenosine monophosphate; and MAP, monoammonium phosphate. Mean and standard error are displayed for $n = 5$ for each treatment.

*$p < 0.05$.

**$p < 0.01$.

***$p < 0.001$.

samples in the middle of the figure; and the samples with AMP and RNA organic treatments on the right side of the first component (Fig. 2).

Pearsons correlation tests, for the control samples, indicated that the microbial community requires more energy to acquire phosphorus than the samples fertilized with other treatments, as shown by the positive correlation between microbial P and the enzymes BG ($r = 0.88$, $p = 0.046$), POX ($r = 0.89$, $p = 0.044$), and Phy ($r = 0.97$, $p = 0.007$; Fig. 2A). These correlations were not observed in the other groups of treatments (the MAP, $Ca(H_2PO_4)_2$, the phytic acid group, and the AMP and RNA group) (Figs. 3B, 3C). However, POX activity was correlated with Phm activity in the AMP and RNA group ($r = 0.89$, $p = 0.004$, Fig. 3C).

In correlations of the cluster of treatments with $Ca(H_2PO_4)_2$, MAP, and phytic acid, the samples showed greater microbial growth when they were able to immobilize more phosphorus, as indicated by the positive correlation of Cmic with Pmic ($r = 0.58$, $p = 0.037$; Fig. 3B). Pmic also correlated negatively with $qCO_2$ ($r = −0.58$, $p = 0.046$; Fig. 3B), which is an indicator of the lower metabolic efficiency of microorganisms when there is insufficient phosphorus within their biomass. However, for this cluster of treatments, Phd, a phosphorus-acquiring enzyme, correlated negatively with Cmic ($r = −0.61$, $p = 0.017$; Fig. 3B).

Finally, in the third group, negative correlations between $qCO_2$ and Pmic were also significant for the AMP and RNA group samples ($r = −0.63$, $p = 0.041$ Fig. 3C), and Nmic and $qCO_2$ presented the same correlation for these treatments ($r = −0.72$, $p = 0.017$). For the same treatment group, a negative correlation was found between DOC and $NO_3^-$ ($r = −0.85$, $p = 0.0035$), and a positive correlation between $NO_3^-$ and $CO_2$-C ($r = 0.8$, $p = 0.04$). The latter also correlated negatively with the NAG enzyme ($r = −0.61$, $p = 0.04$) and with DOC ($r = −0.73$, $p = 0.018$; Fig. 3C).

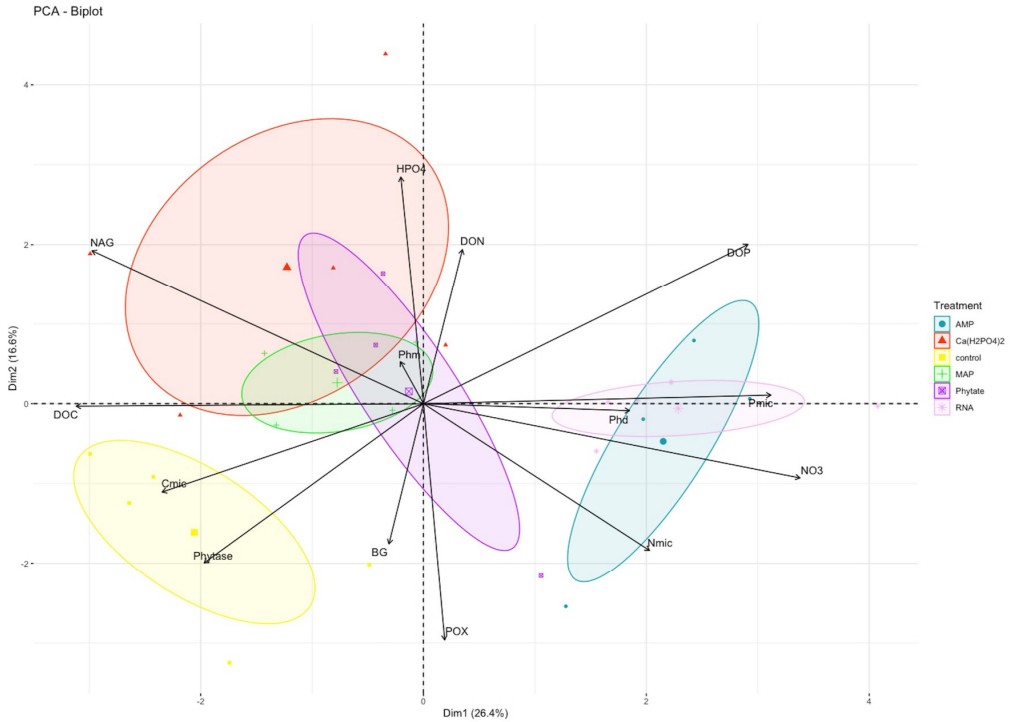

**Figure 2** **PCA analysis for biogeochemical and enzymatic variables obtained after the fertilization incubation experiment.** Each color represents a treatment: Blue for AMP, yellow for Ca(H$_2$PO4)$_2$, green for phytic acid, purple for MAP, pink for RNA. The control is shown in red. AMP, adenosine monophosphate; and MAP, monoammonium phosphate. This figure was made using the "factoextra" package (*Kassambara & Mundt, 2020*) with R software (*R Core Team, 2020*).

## Homeostasis, threshold element ratio, and carbon use efficiency

In most of the treatments, the microbial community is a homeostatic community estimated by a standardized linear regression. A slope that did not differ from 0 ($p > 0.05$) according to the standardized linear regression performed for the control treatments, monobasic ammonium phosphate (MAP), calcium phosphate (Ca(H$_2$PO$_4$)$_2$), RNA, and adenosine monophosphate (AMP) treatments (Figs. 4 and 5) suggest homeostasis. In contrast, the samples treated with phytic acid (phytate) as a source of P comprise a non-homeostatic community, given that in the regressions performed, it exhibited a slope that differed from zero (Figs. 4E and 5E). The microbial community of these soil samples tends to decrease the C:P and C:N ratio of its microbial biomass (immobilization of nutrients) while increasing the C:P and C:N ratio of the resource.

TER$_{C:P}$ analysis showed significant differences between treatments. TER$_{C:P}$ was higher for the samples with the control and MAP treatments, followed by the treatments with Ca(H$_2$PO$_4$)$_2$ and RNA, while the TER$_{C:P}$ was lower for samples with the AMP and phytic acid treatments (Fig. 6). Compared with the dissolved nutrient ratios (DOC:DOP), the TER$_{C:P}$ was found to be lower than this ratio for the control treatments, Ca(H$_2$PO$_4$)$_2$, AMP, RNA, and phytic acid, but the same for the treatment with MAP.

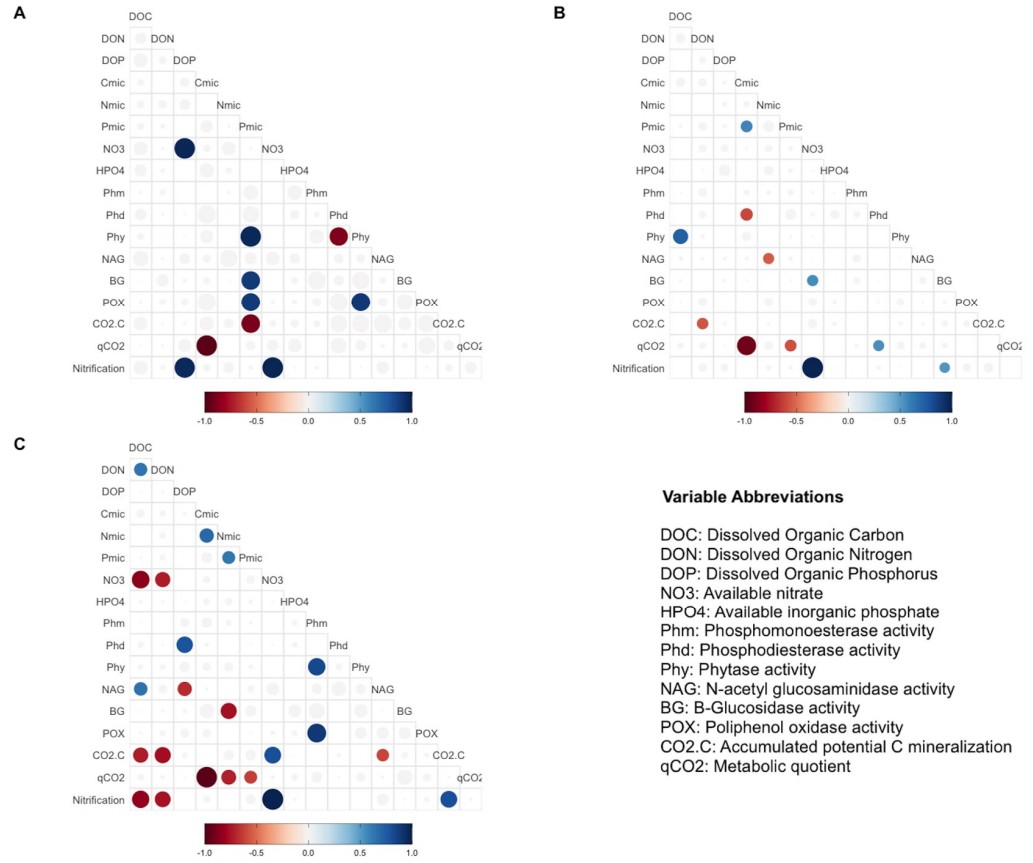

**Figure 3 Pearson correlation test in different treatments group.** Pearson correlation test was performed using biogeochemical and enzymatic variables, C mineralization, $qCO_2$ and nitrification ($DNO_3$) measured after incubation fertilization experiment. The circles represent significant correlations ($p < 0.05$). The color scale indicates the correlation coefficient, and whether the correlation is positive (blue) or negative (red). The correlation analyses are divided by treatment groups according to principal component analyses: (A) Control, (B) MAP, $Ca(H_2PO_4)_3$ and phytic acid group, (C) AMP and RNA group. AMP: adenosine monophosphate, and MAP: monoammonium phosphate. This figure was made using the "ggcorrplot2" package (*Cai & Matheson, 2021*) in R software (*R Core Team, 2020*).

$TER_{C:N}$ was higher for the control, followed in equal measure by the samples treated with AMP, $Ca(H_2PO_4)_2$, MAP, and phytic acid, but lower for the samples treated with RNA (Fig. 7). Compared to the dissolved nutrient ratios, the $TER_{C:N}$ was lower than the DOC:DON ratio for MAP- and RNA-treated soil, while it was higher for the control treatment. For the other treatments (AMP, $Ca(H_2PO_4)_2$, and phytate), the $TER_{C:N}$ was the same as the DOC:DON ratio.

Carbon use efficiency, in relation to phosphorus ($CUE_{C:P}$), did not vary significantly between all treatments (Fig. 8A). However, the value of carbon use efficiency in relation to nitrogen ($CUE_{C:N}$), differ between treatments (Fig. 8B). $CUE_{C:N}$ was higher for the samples treated with $Ca(H_2PO_4)_2$, intermediate for the samples treated with MAP and phytate, and lower for the samples treated with AMP and RNA, and for the control samples (Fig. 8B). The total CUE (Fig. 9) showed a similar trend to $CUE_{C:N}$.

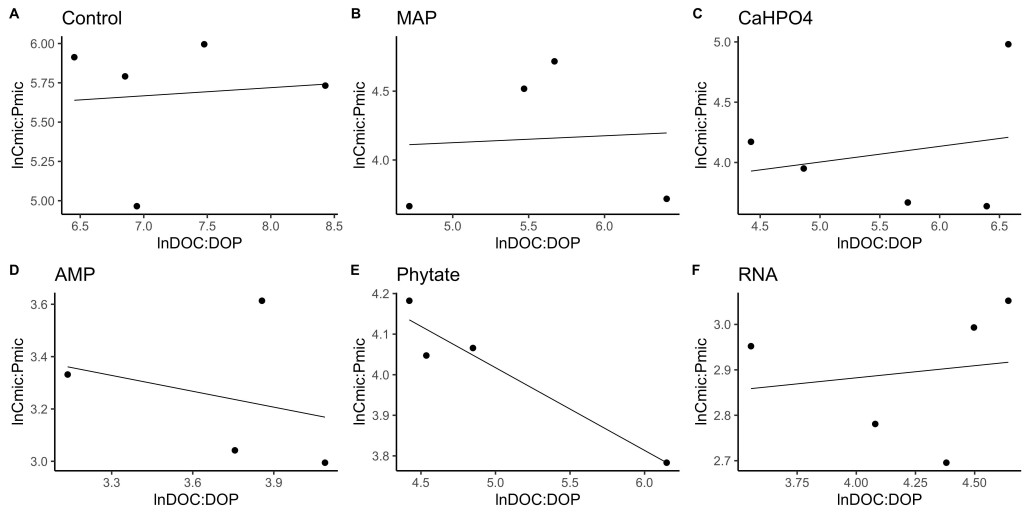

**Figure 4   Soil microbial community homeostasis related to P acquisition estimated by an standardized linear regression.** The treatments are ordered as follows (A) control, treatments: (B) ammonium phosphate (MAP), (C) calcium phosphate Ca(H$_2$PO$_4$)$_3$, (D) adenosine monophosphate (AMP), (E) phytic acid (phytate) and (F) ribonucleic acid (RNA). These values represent strong homeostasis for all treatments because the slope is not different from 0, and there is not a relationship between the microbial biomass quotient and the substrate quotient (DOC:DOP) except for phytate ($p = 0.04$). The equations for each figure are (A) $y = 0.05x + 5.3$, $R^2 = -0.32$. (B) $y = 0.05x + 3.87$, $R^2 = -0.5$. (C) $y = 0.13x + 3.35$, $R^2 = -0.27$. (D) $y = -0.2 + 3.99$, $R^2 = -0.37$. (E) $y = -0.2x + 5.04$, $R^2 = 0.88$. (F) $y = 0.05x + 2.67$, $R^2 = -0.3$.

## DISCUSSION

Soil incubations revealed that the use of specific organic (phytic acid, AMP and RNA) and inorganic phosphate compounds (MAP and calcium phosphate) differently affected nutrient dynamics in soil, such as C mineralization, nitrification, DON and DOP concentrations in soil. Specific enzymatic activity of phosphodiesterase depended on the treatment used.

### Phosphorus sources effect on soil C and N dynamics

All the evaluated phosphorus sources stimulated microbial C mineralization. This suggest that P limits the activity and growth of microbial communities in the selected soil, as was reported in previous studies for the study site (*Perroni et al., 2014*; *Tapia-Torres et al., 2015a*). As hypothesized, labile organic treatments, such as adenosine monophosphate (AMP) and RNA, promoted microbial C mineralization. It has been previously reported that soil bacteria from CCB prefer DNA as a phosphorus substrate over inorganic phosphorus, such as potassium phosphate and calcium phosphate when isolates are grown in culture media (*Tapia-Torres et al., 2016*). The degradation of both DNA and RNA requires phosphodiesterase enzymes, while the substrate AMP can be seen as a monomer from the decomposition of nucleic acids and requires phosphomonoesterase (*Lehninger, Nelson & Cox, 2005*). Both treatments contain not only P but also C and N, which suggests phosphorus colimitations with C and N in the soil, and microbial activity

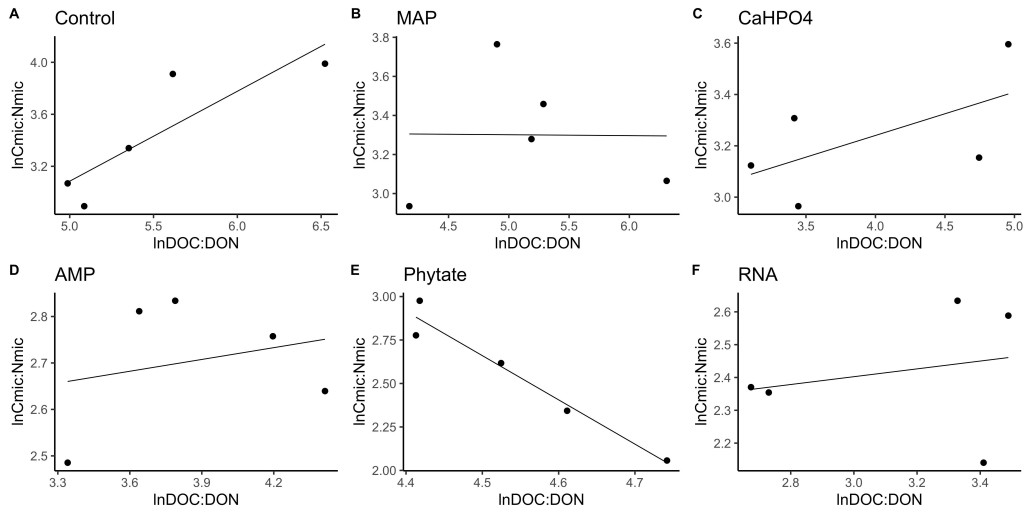

**Figure 5** **Soil microbial community homeostasis related with N acquisition estimated by an standardized linear regression.** The treatments are ordered as follows (A) control, (B) ammonium phosphate (MAP), (C) calcium phosphate $Ca(H_2PO_4)_3$, (D) adenosine monophosphate (AMP), (E) phytic acid (phytate) and (F) ribonucleic acid (RNA). These values represent strong homeostasis for all treatments because the slope is not different from 0, and there is not a relationship between the microbial biomass quotient and the substrate quotient (DOC:DON), except for phytate ($p = 0.04$). The equations for each figure are (A) $y = 0.7x - 0.37$, $R^2 = 0.66$. (B) $y = -0.005x + 3.3$, $R^2 = -0.33$. (C) $y = 0.17x + 2.6$, $R^2 = 0.15$. (D) $y = 0.085 + 2.4$, $R^2 = -0.25$. (E) $y = -2.5x + 14.11$, $R^2 = 0.9$. (F) $y = 0.12x + 2.05$, $R^2 = -0.26$.

is promoted when these nutrients are added. However, the other organic treatment, phytic acid, did not have the same expected effect.

The microbial community and nutrient dynamic response to phytic acid was similar to that for inorganic P substrates, as shown by the principal component analysis and with the accumulated C mineralization results. Moreover, the phytic acid treatment had the highest metabolic quotient ($qCO_2$) value, suggesting that the microbial community is undergoing metabolic stress (*Anderson & Domsch, 1993*) or is a microbial community with high energy requirements (*Carpenter-Boggs, Kennedy & Reganold, 2010*). The metabolic quotient ($qCO_2$) also showed a negative correlation with Pmic in the principal component group of inorganic treatments and the AMP and RNA group, suggesting that metabolic stress has an inverse relationship with the amount of P immobilized in microbial biomass. These results suggest that phytic acid is not a readily available source of P and C for soil microorganisms. Higher energy requirements may be due to phytic acid interactions with soil since it is strongly bound to soil clay and soil organic matter and can react with soil minerals, such as calcium, favoring precipitation and adsorption reactions (*Dalai, 1977*; *Stewart & Tiessen, 1987*; *McKercher & Anderson, 1989*; *Wan et al., 2016*) and becoming less susceptible to microbial attack. This is important in the CCB soils given their high sorption capacity, which is greater where higher concentrations of organic compounds are found (*Perroni et al., 2014*). These tend to be higher in agricultural fields than in natural soils, because the continuous water and nutrient inputs increase total organic carbon and

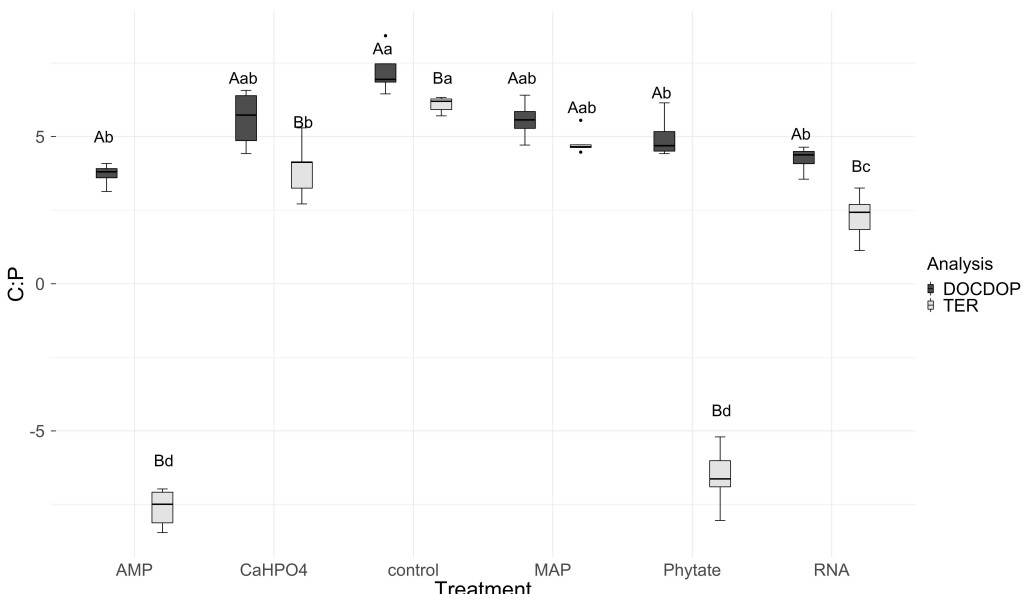

**Figure 6  Mean of natural logarithms of DOC:DOP ratio and TER$_{C:P}$ of all treatments.**  Significant differences for the comparisons between DOC:DOP ratio (DOCDOP, black boxes) and the TER$_{C:P}$ (gray boxes) values of each treatment are marked with uppercase letters, while the significant differences of the TER$_{C:P}$ or DOC:DOP values between treatments are marked with lowercase letters.

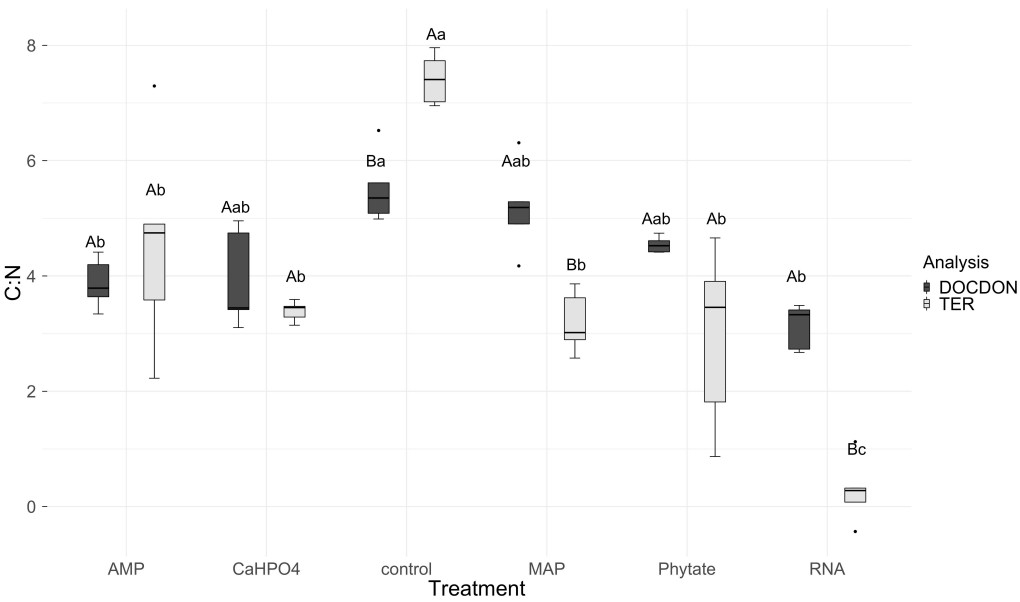

**Figure 7  Means of natural logarithms of DOC:DON ratio and TER$_{C:N}$ of all treatments.**  Significant differences for the comparisons between DOC:DON ratio (DOCDON, black boxes) and the TER$_{C:N}$ (gray boxes) values of each treatment are marked with uppercase letters, while the significant differences of the TER$_{C:N}$ or DOC:DON values between treatments are marked with lowercase letters.

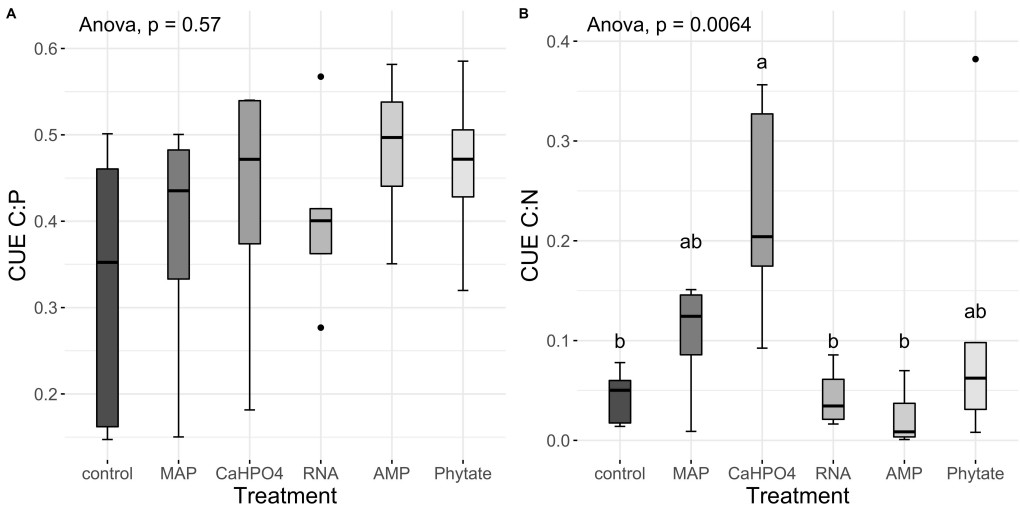

**Figure 8  Carbon use efficiency related to P and N.** Means for (A) $CUE_{C:P}$ y (B) $CUE_{C:N}$. Letters show significant differences between treatments obtained with the Tukey HSD test. The *p* value from the ANOVA analysis is shown on top of the figures.

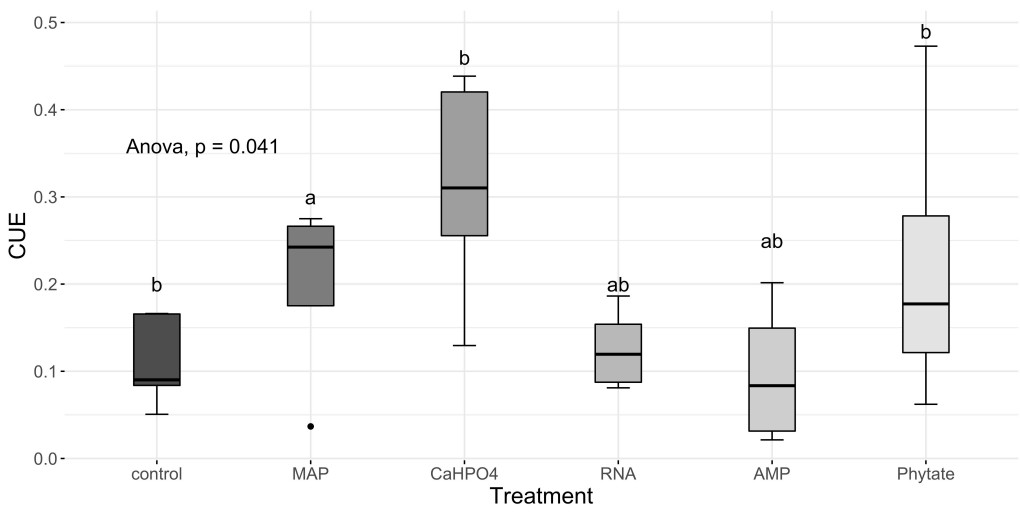

**Figure 9  Means of CUE calculated with Eq. (5).** Letters show significant differences between treatments obtained with an LSD test. The *p* value from the ANOVA analysis is shown on top of the figure.

dissolved organic phosphorus in soils, compared to that of native grasslands (*Hernández-Becerra et al., 2016*). As a consequence, phosphorus acquisition from phytic acid molecules is a two-step process, which demands more energy from soil microorganisms. First, insoluble and mineral-bound phytate compounds must be solubilized by bacteria or fungi capable of synthesizing organic acids and chelates (*Hill & Richardson, 2007*). Free and soluble phytic acid can then be hydrolyzed by phytases (*Lim et al., 2007*); specifically, B-propeller phytase, the active phytase type in neutral and alkaline soils, which breaks

down each bound monoester to release inorganic phosphate (*Gontia-Mishra & Tiwari, 2013*; *Cotta et al., 2016*). Only after the complete dephosphorylation of the molecule is phytic acid transformed into *myo*-inositol, which can then be used as a carbon source by soil microbes (*Cosgrove, Irving & Bromfield, 1970*) as it has been reported for some bacteria (*Chen et al., 2020*; *Rothhardt et al., 2014*; *Yuan et al., 2019*). These results suggest that the molecular structure plays an important role in its decomposition, rather than simply the concentration of C, N, or P.

Addition of labile organic molecules (AMP and RNA) can also affect soil N dynamics, promoting nitrification rates and thus increasing the susceptibility to soil N losses (*Tapia-Torres et al., 2015b*). Two processes can explain this result. The first addition of labile organic molecules could prime the microbial community (*Garcia et al., 2017*; *Scotti et al., 2015a*; *Scotti et al., 2015b*). In our case, the organic treatments AMP and RNA acted as these OM inputs and served as an initial energy source for microorganisms capable of mineralizing soil OM. The measured DOC concentration was therefore lower in the AMP and RNA treatments at the end of the experiment, probably because of the high rate of depletion of C sources, as well as Cmic for both treatments, while the $Ca(H_2PO_4)_2$ treatment contained the highest concentration of DOC at the end of the experiment. These findings suggest that lower DOC results from higher C mineralization rates since it is negatively correlated, producing the depletion of labile organic matter in the soil. As a consequence, the carbon use efficiency, in relation to nitrogen ($CUE_{C:N}$), was lower in the AMP and RNA treatments than with monobasic calcium phosphate ($Ca(H_2PO_4)_2$). These CUE results can be explained by the biogeochemical analysis performed at the end of the incubation period when the DOC was consumed by the microbial community. $CUE_{C:N}$ is expected to decrease when C is a limiting resource and the remaining organic matter for decomposition has higher recalcitrance (*Sinsabaugh et al., 2013*). Second, a decrease in DOC at the end of the incubation are related to an increase of the activity of nitrifier. Our results showed that the DOC concentration correlated inversely with nitrate in both of these treatments, which suggests enhanced nitrifier activity. These chemoautotrophs obtain energy from oxidizing $NH_4$ to $NO_2^-$ and $NO_2^-$ to $NO_3^-$ (*Fenchel et al., 2012*). Moreover, these bacteria present optimum activity in a neutral to alkaline pH (*Prosser, 1990*) and nitrification rates increase with higher pH (*Li et al., 2020*), which is coincident with the soil studied herein. A decrease of COD at the end of the incubation in these soils could make these bacteria increase and be competitive with the heterotrophic bacteria. In a pulse of carbon, such as that created by the application of organic treatments AMP and RNA, rapidly growing heterotrophic (r strategist) microbes immobilize nutrients and grow faster, which may occur during the first days of the incubation. However, enhanced growth of these organisms may induce a rapid depletion of labile carbon sources, giving place to a reduction of r strategist bacteria, and an increase in k strategist and chemoautotrophic bacteria (*Montaño Arias & Sánchez-Yañez, 2014*), which also explains the reductions in microbial C. A decrease of NOD in the AMP treatment while $NO_3^-$ increases is an indicator that heterotrophic bacteria are mineralizing organic matter containing N, and yielding $NH_4^-$ as a result of organic C limitation (*Chapin III, Matson & Vitousek, 2011*). The $NH_4^+$ is then rapidly used as a substrate for the nitrifiers.

These two processes suggest that, while organic labile substrates such as ARN and AMP may favor microbial respiration, it could be important to consider a constant supply of organic amendments in agricultural practices to avoid soil N losses.

## Effect of phosphorus addition on P availability

In this study, we hypothesized that AMP and RNA treatments would promote the availability of soil nutrients, particularly phosphorus. However, we did not find an increase in concentrations of available $HPO_4^{-3}$ at the end of incubation, but instead found higher DOP and Pmic concentrations in both labile organic treatments (RNA and AMP). Increases in Pmic are crucial because the microbial community is retaining labile forms of P in actively cycling biological pools, and reducing the rate at which labile inorganic P would otherwise be permanently lost *via* adsorption onto soil particles or leaching (*Cleveland, Townsend & Schmidt, 2002*). On the other hand, organic phosphorus compounds are an essential fraction of the total P in the soil since, in the CCB grasslands, they can represent about 50% of the total P (*Perroni et al., 2014*), and dissolved organic phosphorus is composed principally of products of microbial metabolism (*Cleveland, Townsend & Schmidt, 2002*).

Besides the changes in organic and microbial P pools, the specific enzyme activity (SEA) of the phosphorus enzymes differed among treatments. Organic treatments, whether AMP, RNA, or phytic acid, stimulated phosphodiesterase activity per unit of microbial biomass, as shown with the SEA of Phd, whereas the phosphomonoesterase enzyme was unaffected. Phosphomonoesterases and phosphodiesterases are parts of the phosphate regulon (Pho) in bacteria, which is responsible for phosphorus uptake and responds to P starvation (*Santos-Beneit, 2015*). Lower concentration of inorganic phosphate but higher availability of organic P in the organic treatments at the beginning of the experiment may have enabled the production of the Phd enzyme. Extracellular enzymes can persist in soil, associated with clay and organic matter particles, and remain active (*Nannipieri et al., 2011*). It is therefore possible that Phd could have persisted until the end of the experiment.

Nucleic acids, such as RNA and DNA, are released by dead cells in the environment and constitute an important labile source of nutrients such as C, N, and P (*Tani & Nasu, 2010*), particularly for bacteria from oligotrophic environments (*Tapia-Torres et al., 2016*). In CCB soils Phd activity tends to be higher than Phm activity (*Tapia-Torres et al., 2016*; *Montiel-González et al., 2017*), demonstrating that phosphodiester uptake plays a prominent role in phosphorus cycling in these soils (*Tapia-Torres et al., 2016*). These studies agree with *Turner & Haygarth (2005)*, who determined, in pasture soils, that phosphodiesterase activity is the rate-limiting step that regulates P turnover because P availability depends on the degradation of fresh organic materials, which are abundant in phospholipids and nucleic acids, cellular components that are sources of phosphate diesters.

Phosphorus turnover is highly important in agricultural systems because inorganic phosphorus tends to be lost or become unavailable to crops due to lixiviation or occlusion processes. Although inorganic P is the immediate source of P for vegetation, it is necessary to promote an increase in labile organic P molecules and microbial P pools to prevent these losses, and an increase in the enzymes that hydrolyze organic P compounds, such
as phosphomonoesterases, phosphodiesterases, phytases, and phosphonatases, to allow a slow but constant release of inorganic P.

## Effect of phosphorus addition on C, N, and P stoichiometry

In most treatments, the microbial community was homeostatic, *i.e.,* the C:N:P ratios in the microbial biomass remained constant despite changes in these ratios in the resources (*Elser & Sterner, 2002*). Nevertheless, the microbial community in the phytic acid treatment was non-homeostatic. A common premise used in ecological stoichiometry studies is that heterotrophic organisms are strictly homeostatic, while autotrophs can present a changing stoichiometry (*Persson et al., 2010*; *Fanin et al., 2013*), although there are some scenarios in which members of a microbial community can change their stoichiometry according to that of their resource, thus becoming non-homeostatic. Non-homeostatic behavior is a mechanism by which to reduce stochiometric imbalances between the resources and microbial biomass (*Mooshammer et al., 2014*) because it can occur through the microbial storage of nutrients in excess or by shifts in microbial community structure and therefore shifts in the biomass stoichiometry (*Mooshammer et al., 2014*). We reported a lower value of the Cmic:Pmic ratio compared to that of the control, suggesting greater P immobilization with P addition; however, this difference was present in all phosphorus treatments, not just that of phytic acid. *Fanin et al. (2013)* suggested that non-homeostatic behaviors are the result of changes in microbial community composition rather than shifts in the microbial biomass of individual microorganisms since they found that the bacteria:fungi and gram positive:gram negative ratios change along with changes in homeostasis. For example, the reported fungal C:N:P ratio is 250:16:1 (*Zhang & Elser, 2017*), while the bacterial C:N:P ratio is 46:7:1 (*Cleveland & Liptzin, 2007*). In the phytic acid treatment, the average C:N:P ratio was 58:5:1 and thus closer to the bacterial biomass ratio, or the average soil microbial biomass ratio (60:7:1) suggested by *Cleveland & Liptzin (2007)*. However, in the pre-incubation samples, the microbial biomass stoichiometry was closer to the fungal biomass stoichiometry (246:16:1, Table 1), as was the case in the control samples (310:10:1; Table 4). This suggests that the homeostasis imbalances are due to the microbial community changing to different microbial groups with the addition of fertilizers. Regarding the phytic acid treatment, bacteria are the main producers of B-propeller phytase, the active phytase type in neutral and alkaline soils, while fungi are the producers of acid phytases (*Jain, Sapna & Singh, 2016*).

The threshold element ratio (TER) is the elemental proportion that corresponds to balanced microbial growth, with no limitation by C or nutrients (*Sinsabaugh et al., 2016*). It represents the critical ratio at which organisms transition from net nutrient immobilization to net nutrient mineralization (*Mooshammer et al., 2014*) and it defines whether the community is limited by nutrients (N or P) or by energy (C). When resource C:N or C:P ratios are greater than the TER, the system is limited by nutrients and immobilization processes dominate. However, when these resource ratios are lower than the TER, then the system is limited by energy, and nutrient mineralization occurs (*Sinsabaugh et al., 2013*). In this study we selected the treatment MAP because it was used as a fertilizer in the agricultural plots from which the soil was obtained. This treatment did not show

differences between the DOC:DOP ratio and TER and it can therefore be considered that the soil microbial community is co-limited by phosphorus and energy (*Sterner & Elser, 2002*). In contrast, for the $Ca(H_2PO_4)_2$, AMP, RNA, and phytic acid treatments, there is a limitation by P for the soil microbial community because TER was lower than the soil DOC:DOP ratios. In this case, the microbial community is inclined to immobilize available phosphorus. This coincides with previous studies conducted with non-managed soils from CCB, at the eastern side of the valley (Pozas Azules), where low concentrations of DOC, a trend for phosphorus limitation, and low values of $TER_{C:P}$ were found (*Tapia-Torres et al., 2015a*).

Regarding nitrogen, the $TER_{C:N}$ was lower than the DOC:DON ratio for MAP and RNA-treated soil, suggesting that the microbial community is limited by N, which indicates a tendency to immobilize N, although this is not shown in the microbial biomass. It was also found that, in a CCB site from the western side of the valley (Churince) with higher DOC values as well as in our MAP treatment, there was a limitation by N (*Tapia-Torres et al., 2015a*). On the other hand, the $TER_{C:N}$ was higher for the control treatment, implying limitation by C or energy and a preference for mineralizing organic nitrogen compounds to obtain C and release $NH_4^+$ while immobilizing more C and reducing its losses through mineralization. The $CUE_{C:N}$ discussed previously ('Phosphorus sources effect on soil C and N dynamics') reflected this carbon limitation since the AMP and RNA treatments had the lowest CUE. These results all suggest that the addition of labile organic molecules with P (MAP and RNA) acts to increase microbial N limitation, probably through the increased demand for N by the growing microbial community brought about by the priming effect discussed previously ('Phosphorus sources effect on soil C and N dynamics').

The results of this study show that the soil microbial community responds differently to different phosphorous molecules. These effects show differences both between organic and inorganic molecules and among the same groups of molecules with different chemical compositions. This can have implications when conducting fertilization with organic matter in field crops since the chemical structures of the molecules that make up composts and manures are usually unknown. Although the most labile organic compounds (AMP and RNA) favored C mineralization, they also showed a rapid decrease in DOC, implying a reduction in microbial biomass and an increase in chemoautotrophic microorganisms such as nitrifying bacteria, indicating that when fertilizing with labile organic sources, the periodicity of application must be carefully considered to avoid soil N losses.

## CONCLUSIONS

Despite having carried out this experiment using soil from an agricultural field with conventional management, the soil microorganisms showed P and C limitations. Such C limitations and low CUE levels indicate highly recalcitrant soil C compounds, and this is also reflected in the microbial biomass ratios, which were similar to soil fungi biomass ratios. Carbon limitations were overcome with phosphorus fertilization and P treatments promoted the immobilization of this nutrient in microbial biomass and, in some treatments (AMP, RNA, and $Ca(H_2PO_4)_2$), promoted the increase in DOP. All fertilizations reduced

the soil microbial biomass ratios, which could be an indicator of a changing microbial community and an increase in bacterial biomass relative to fungal biomass. Although P in microbial biomass might not be available to crop plants immediately, it is an organic phosphorus pool that is quickly recycled and can protect P from losses through leaching and adsorption to soil minerals. The labile organic treatments (AMP and RNA) increased the availability of N, although this nutrient was quickly nitrified. Nitrate is a form of N that is available to plants, but it is susceptible to loss from the soil.

## ACKNOWLEDGEMENTS

We thank the reviewers for comments on a draft of the manuscript. This paper is presented by Pamela Chávez-Ortiz as partial fulfillment of a doctoral degree at the ''Programa de Posgrado en Ciencias Biologicas, UNAM''. We also thank Rodrigo Velazquez-Durán for their assistance during chemical analyses.

### Funding

This research was funded by the PAPITT-DGAPA, UNAM (IN207721). The ''Consejo Nacional de Humanidades Ciencia y Tecnología'' scholarship was provided to Chávez-Ortiz during her doctoral studies (CONAHCYT No. CVU 630699). The funders had no role in study design, data collection and analysis, decision to publish, or preparation of the manuscript.

### Grant Disclosures

The following grant information was disclosed by the authors:
PAPITT-DGAPA, UNAM: IN207721.
Consejo Nacional de Humanidades Ciencia y Tecnología: CVU 630699.

### Competing Interests

The authors declare there are no competing interests.

### Author Contributions

- Pamela Chavez-Ortiz conceived and designed the experiments, performed the experiments, analyzed the data, prepared figures and/or tables, and approved the final draft.
- John Larsen conceived and designed the experiments, authored or reviewed drafts of the article, and approved the final draft.
- Gabriela Olmedo-Alvarez conceived and designed the experiments, authored or reviewed drafts of the article, and approved the final draft.
- Felipe García-Oliva conceived and designed the experiments, performed the experiments, analyzed the data, authored or reviewed drafts of the article, and approved the final draft.

### Data Availability

 The raw data are available in the Supplementary File.

## Supplemental Information

Supplemental information for this article can be found online at http://dx.doi.org/10.7717/peerj.18140#supplemental-information.

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
