# Peer review of "Control of inorganic and organic phosphorus molecules on microbial activity, and the stoichiometry of nutrient cycling in soils in an arid, agricultural ecosystem"

_PeerJ, doi:10.7717/peerj.18140_

## Round 0.1 · original submission · Major Revisions

Two reviewers carefully and constructively reviewed this submission. They both see value in this study. I agree with them and recommend you resubmit after their detailed suggestions are addressed. I have also made suggestions for format and style (see below). These are not comprehensive. I gave that up after line 100 and picked a few examples from the rest of the draft.

Regards,

Michael

I am not sure of the title. Does this work? “Control of inorganic and organic phosphorus molecules on microbial activity, and the stoichiometry of nutrient cycling in soils in an arid, agricultural ecosystem”
Line 21. Background is misspelled.
Start with a statement about why we need to understand biogeochemical cycling in microbial activity in arid soils. Does it relate to agriculture, climate change…. You mention agriculture in the conclusions, so perhaps that is a justification for this work.
Line 23. Text is wordy. Revise to “..impact the ability of soil microbes to produce the enzymes required to make nutrients available.”

Line 26. Conducted with what soils? Provide some context (alfalfa fields in Mexico).

Line 30 “after 19 days”

Line 31. Here and throughout, avoid the passive voice. Revise to “The type of P affected potential…”

Line 34. What organic compounds? Soils amended with organic P?

Line 35. Revise to “…organic carbon and increased nitrification.

Line 46. Delete “for the metabolism of living organisms”

Line 47. Revise to “…biomolecules, including…”

Line 48. Revise to “In soil, P mainly originates from the weathering of apatite (Schlesinger…”

Line 50. Delete “The”

Line 56. Delete “followed…phospholipids”

Line 68. Revise to “to the availability of C (Luo et al., 2019), N and other elements (Olander & Vitousek, 2000).”

Line 80. Revise to “TER analyses have been reported for natural terrestrial ecosystems (Tapia-Torres et al., 2015a; Montiel- González et al., 2017; Cui et al., 2018a; Cui et al., 2018b) and
managed ecosystems (Zhang et al., 2020) but ___ studies have anlayzed TER in agricultural systems.”

Line 90. Revise to “Carbon Use Efficiency (CUE) represents the efficiency with which bacterial populations convert organic carbon substrates into biomass and is quantified…”

Line 93-97. Revise to “CUE corresponds to the rate microbial communities decompose organic matter and release CO2 and is a function of the ability…” Note, don’t call out an author and say they suggested something, unless you doubt their work. Write the science. Cite the scientists.

Line 301 - 303. Replace (if appropriate). “With ..simple …” with “TER was calculated by performing simple …”

Line 333. Replace “(CUE) can be measured from ecological stoichiometry. In this study, we calculated..” with “(CUE) was calculated…”

Line 382. Do not start a paragraph in Results by sending the reader away to check out the data. Start each paragraph with a statement of the key result (Organic P treated soils showed high… (Table 2). See also lines 436, 466

Line 394. Delete “The nutrient availability analyses demonstrated that the”

Line 402. Revise to “Both organic and inorganic P treatments increased DOP…”

Line 416. Revise to “Enzyme activity was only significantly…”

Line 421. Delete “Our experiments showed that the” and avoid phrases like that in Results. Are you presenting someone else’s results?

Line 428. Delete “Regarding NAG SEA values, the” and similar fillers throughout.

Line 433. Save interpretation of results (“suggests that”) for Discussion.

Line 447. Move to Methods. See also 457.

Line 486. Revise to (Fig. 4 and 5).

Line 514 -520. Revise this section to start with a topic sentence that presents the key finding of this study (“Soil incubations revealed/confirmed that P limit microbial activity…”).

Line 728. Replace “are indicators” with “indicate”

Line 755. pH

Line 766. Only cap proper nouns (not “Biodynamic”). See also 776, 784

Line 797. Extra period.

Fig 6 - 9. Present box plots not bars.

Reviewer 1 ·

Basic reporting

The manuscript titled "How inorganic and organic phosphorus molecules modify microbial activity, stoichiometry, and nutrient dynamics in an agricultural soil of an arid ecosystem" delves into a compelling and impactful research area. The introduction offers a comprehensive overview, delving into the necessary background and literature review. The Materials and Methods section is meticulously detailed, providing clarity on the experimental design. However, in certain sections of the Results and Discussion, the coherence of the writing or the focus of the research investigation seems to waver. Some figures require attention, and certain statistical analyses yield non-promising results due to P-values exceeding 0.05. Additionally, the rationale behind the inclusion of certain parameters lacks clarity and logical reasoning.

Experimental design

An important issue with the experimental research is the absence of information regarding the soil type utilized for this study (lines172-178). Furthermore, the water holding capacity during the incubation experiment reached 90% (line 200), resembling conditions close to waterlogging. To draw conclusive findings, the experimental design should incorporate multiple soil types instead of just one and be aware of the aerobic condition of the soil environment. The lack of sampling diversity, and appropriate environment of microbial activities will diminish the study's significance.

Validity of the findings

The statistical analysis appears questionable due to the higher p-values observed in Figure 6 and 7. Figure 5 requires attention regarding the absence of digits on the X-axis (Fig 5F).

Additional comments

This manuscript requires significant revisions in data analysis. Additionally, the authors must address flaws in their experimental design and clearly specify whether it is an aerobic or anaerobic study.

Reviewer 2 ·

Basic reporting

I suggest rewording the introduction to match the hypothesis. The current version is conceptually incomplete and does not facilitate the reader's understanding because different P forms can modify microbial activity. How do the authors expect the P acquisition resources invested through specific mechanisms to impact carbon use efficiency (i.e., depolymerization for RNA, solubilization for calcium phosphate, mineralization for adenosine monophosphate, etc.)?
As the authors refer, I suggest that redaction improvement includes the design's dependency on the soils employed.
It is worth noting that only 3 of 33 bibliographies used be from the last ten years.
I suggest including the TER, CUE, and SEA responses in the title. Maybe "Respon" e of TER, CUE, and SEA to different P sources agricultural soil of an arid oligotrophic ecosystem.

L1-3 I suggest modifying the title because, formally, this article refers to "how much" and not "how" because the authors evaluate how much they have changed the metabolic indicators due to the addition of different forms of phosphorus.
L26-30 Add a brief description of the soil used for the experiment (i.e., arable layer of calcareous soil)
L26 Modify to "…molecules as treatment substrates".
L46 Eliminate "live" because it is not necessary. After all, dead organisms do not have an active metabolism.
L47-48 Add information on metabolic regulation and trade-offs under P-limitation conditions.
L48 Add mainly to the beginning of the sentence.
L49 Moving apatite to the end of sentences is an example of one of many.
L54 Avoid very short sentences.
L57 Improve sentence: Organic phosphate molecules - and phosphates released by mineralization - can be made available to plants and soil microorganisms through the action of secreted enzymes (exoenzymes) produced by soil microorganisms.
L69, what other elements? Because they can remove nutrients from potentially toxic elements through different mechanisms.
L108 The correct nomenclature of isotopes is with the mass number to the element's left as 13C.
L110 correlated positive or negative? What is this importance in the context of the substrates used in this study?
L111-112 Commonly, the presence of proteins in 13C NMR is related to carboxylic and N-O-alkyl signals, including methoxyls.
L148-155 The conceptual form presented above is weakly linked to the hypothesis; this makes it challenging to identify the coherence between the theoretical premise and the methods used.

Experimental design

No comment

Validity of the findings

The findings lose relevance and impact without a well-executed introduction, although they are rationally sound and novel.

Additional comments

No comment

---

## Round 0.2 · Minor Revisions

I have attached an annotated pdf with specific changes I suggest. Please consider these suggestions and the comments from the reviewer.

Regards,

Michael

Reviewer 1 ·

Basic reporting

The manuscript, "Control of Inorganic and Organic Phosphorus Molecules on Microbial Activity and the Stoichiometry of Nutrient Cycling in Soils in an Arid Agricultural Ecosystem," illustrates how various forms of phosphorus (P) in organic and inorganic fertilizers affect the ability of soil microbes to produce the enzymes necessary for releasing nutrients such as carbon (C), nitrogen (N), and phosphorus (P) from different substrates, thus impacting biogeochemical cycles. While the manuscript is well-written and includes a comprehensive literature review, robust experimental design, and effective data visualization and analysis, improvements could be made by incorporating more complex sentences, shortening the introduction, and ensuring consistent use of abbreviations.

Experimental design

The experimental design is thoroughly explained, though some paragraphs (for example, line 198-211, 226-235) are quite brief and could benefit from consolidation. The research questions are addressed systematically throughout the methodology. While sufficient details are provided, equations (for example, line 273, 275, 295, 300, 307) should be formatted appropriately for clarity.

Validity of the findings

The results section is promising, featuring clear data visualization and thorough statistical analysis. The findings are presented in a coherent, step-by-step format.

Additional comments

Key revisions include: 1. Shorten the introduction, 2. Combine brief paragraphs, and 3. Review and standardize abbreviation formatting.

Reviewer 2 ·

Basic reporting

The document is improved by the authors after attending the reviewer's comments.

Experimental design

The experimental design and data analyses are adequate and robust.

Validity of the findings

The findings are valid and exciting when they are not entirely novel.

Additional comments

The current introduction is adequate, but the references are weak because many of them are self-citations or do not reflect the trend in this paper's investigation scope over the last five years.

---

## Round 0.3 · Minor Revisions

I think manuscript only needs the following changes. I can then accept without further review.

Line 54. Don’t delete the entire sentence “Through the production …in the soil..” Just delete “Through the production” and start paragraph with “Microorganisms.."

Line 357. As stated in the last review, do not tell reader to go look at a data (“Descriptive soil… table2”). Describe what you what is important in that data and then direct the data to it. Also, “T” in table is cap.

Line s 434, 435 and 438. “F” in Fig. should be cap. Check this throughout.

Line 475. Revise to “This suggests that P limit the activity..”

Line 478. Add commas around phrase “…, such as …and RNA,”

Line 616. Revise to “bacteria are the main producers”

---

## Round 0.4 · accepted · Accept

I did notice one typo that should be addressed in production:
line 465. Revise to "This suggests that P limits the activity and growth...